# STITCH: TRAINING-FREE POSITION CONTROL IN MULTIMODAL DIFFUSION TRANSFORMERS

## ABSTRACT

Text-to-Image (T2I) generation models have advanced rapidly in recent years, but accurately capturing spatial relationships like "above" or "to the right of" poses a persistent challenge. Earlier methods improved spatial relationship following with external position control. However, as architectures evolved to enhance image quality, these techniques became incompatible with modern models. We propose Stitch, a training-free method for incorporating external position control into Multi-Modal Diffusion Transformers (MMDiT) via automatically-generated bounding boxes. Stitch produces images that are both spatially accurate and visually appealing by generating individual objects within designated bounding boxes and seamlessly stitching them together. We find that targeted attention heads capture the information necessary to isolate and cut out individual objects mid-generation, without needing to fully complete the image. We evaluate Stitch on PosEval, our benchmark for position-based T2I generation. Featuring five new tasks that extend the concept of *Position* beyond the basic GenEval task, PosEval demonstrates that even top models still have significant room for improvement in position-based generation. Tested on Qwen-Image, FLUX, and SD3.5, Stitch consistently enhances base models, even improving FLUX by $218\%$ on GenEval's *Position* task and by $206\%$ on PosEval. Stitch achieves state-of-the-art results with Qwen-Image on PosEval, improving over previous models by $54\%$, all accomplished while integrating position control into leading models training-free.

## 1 INTRODUCTION

Text-to-Image (T2I) generation models provide a powerful bridge between natural language and visual content. By transforming text descriptions into images, they expand human creativity, accelerate prototyping, and enable a wide range of applications (Chen et al., 2023; Zhou & Lee, 2024). But as their popularity grows, so do user demands. These models should be able to generate real-world scenes with uncommon but still physically plausible object arrangements. They should interpret and fulfill requests that are both complex and linguistically nuanced in their descriptions of positioning. All of these needs could be met with strong position understanding, yet even basic spatial concepts such as "above" or "to the right of" remain a persistent challenge for current models (Ghosh et al., 2023) (Huang et al., 2023), and the difficulty only grows with more complex prompts.

Luckily, positional performance can be improved by augmenting off-the-shelf models with bounding boxes generated by Large Language Models (LLMs) (Fang et al., 2025; Feng et al., 2023; Yang et al., 2024; Lian et al., 2024). But while many older models such as SDXL (Podell et al., 2023) were built on U-Net architectures, newer models like FLUX (BlackForest, 2024) instead adopt transformer-based designs (Esser et al., 2024). In U-Net-based architectures, cross-attention between the image and the prompt embeddings occurs at a single, well-defined place, whereas in transformer-based architectures it is distributed across many layers. Most bounding-box methods intervene on the cross-attention, but naively extending this across all layers leads to a mismatch in constraint strength: applied too weakly, the model ignores the layout, and applied too strongly, it introduces visible seams. Although some target this adaptation challenge (Chen et al., 2025c; 2024a), achieving maximum control without losing coherence remains far from solved.

To bridge this gap, we propose Stitch: a test-time technique that effectively improves positional prompt generation (Figure 1a) by integrating additional position support via LLM-generated bound-

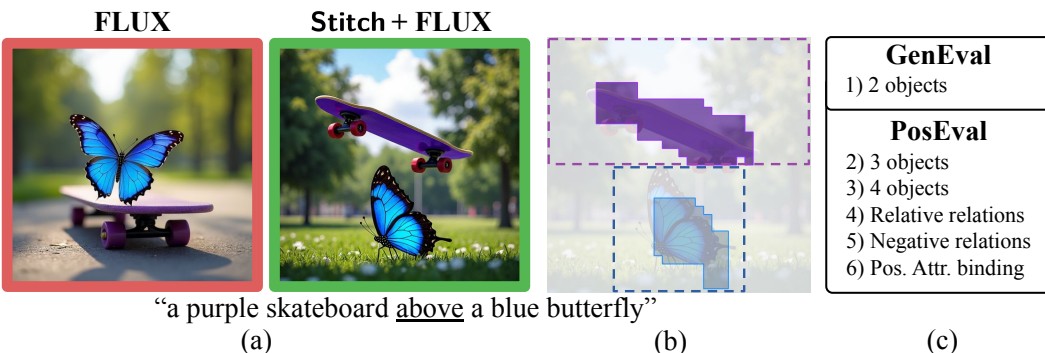

**FLUX** **Stitch + FLUX**

"a purple skateboard above a blue butterfly"
(a)                                    (b)                                    (c)

**GenEval**
1) 2 objects

**PosEval**
2) 3 objects
3) 4 objects
4) Relative relations
5) Negative relations
6) Pos. Attr. binding

Figure 1: (a) **Stitch** boosts position-aware generation, training-free, (b) by generating objects in LLM-made bounding boxes (dashed lines) and using attention heads for tighter latent segmentation mid-generation (filled). (c) Our **PosEval** benchmark extends GenEval with 5 new positional tasks.

ing boxes into leading Flow Matching (FM) T2I models based on Multi-Modal Diffusion Transformer (MMDiT) architecture. Stitch achieves controlled generation by independently generating objects for $S$ steps, with each object constrained to its respective bounding box through attention modulation. Next, we extract foreground objects and combine the predictions. In particular, we find that foreground masks can be inexpensively derived directly from the attention heads (Figure 1b), on-the-fly before completing generation and without requiring an external model. After step $S$ all constraints are lifted, enabling the T2I model to refine the image organically in the remaining steps. Consequently, Stitch facilitates quick and affordable upgrades in the positional performance of the leading T2I models, combining the best image quality with strong positional generation (Figure 2).

Such positional upgrades are extremely valuable for improving 2D spatial performance. Because while recent successes on *Position* tasks in existing benchmarks such as GenEval (Ghosh et al., 2023) might suggest that the problem is nearing resolution, a closer analysis shows that it is far from solved. We take the next step in evaluating whether T2I models can consistently and robustly generate images that accurately reflect positional prompts. To this end, we introduce PosEval, an extension of the GenEval (Ghosh et al., 2023) benchmark designed for in-depth evaluation of positional abilities in T2I generation, going beyond the traditional *Position* category (Figure 1c). PosEval includes five new tasks, each aimed at probing specific failure modes in T2I models. Using PosEval, we demonstrate that state-of-the-art (SOTA) models continue to struggle with positional tasks, highlighting considerable room for improvement. PosEval additionally provides a comprehensive evaluation platform that rigorously measures Stitch's effectiveness in improving positional generation, showcasing significant performance improvements compared to baseline models.

The primary contributions of this work are: (1) We introduce Stitch, a test-time method that substantially improves MMDiT-based models' capacity to accurately generate images from position-based prompts; (2) We find that certain attention heads encode sufficient information to extract the foreground object from the background within the latent space, long before the image is fully generated; and (3) We present the PosEval benchmark, a GenEval (Ghosh et al., 2023) extension featuring five new targeted, position-focused tasks designed to tackle the next level of generation challenges.

## 2 RELATED WORKS

**T2I generation.** T2I synthesis has rapidly advanced in the past years (Podell et al., 2023; OpenAI, 2023; Midjourney, 2025; OpenAI, 2024; DeepMind, 2025), moving from diffusion-based models like Stable Diffusion (Rombach et al., 2022), Imagen (Saharia et al., 2022), and DALL·E (OpenAI, 2023) to more powerful FM approaches like FLUX (BlackForest, 2024), SD3.5 (Esser et al., 2024), and HiDream (Cai et al., 2025), advancing fidelity, diversity, and controllability. They enhance creativity, productivity, and communication (Chen et al., 2023; Zhou & Lee, 2024; Zhou et al., 2024). Previous works have improved pre-trained T2I models by enhancing prompt adherence (Eyring et al., 2024; 2025), alignment with human preferences (Karthik et al., 2025; Liang et al., 2024; Bravo, 2025; Sendera et al., 2025), and personalization (Liu et al., 2023b; Ruiz et al., 2022; Kim

"A photo of a house with a tree to the left and a white horse is on the same side of the house as the tree and below it. The orange cat is not to the left of the house."

## Linguistic Challenges

### Unusual Scenes

"A rainbow beneath a dinosaur"

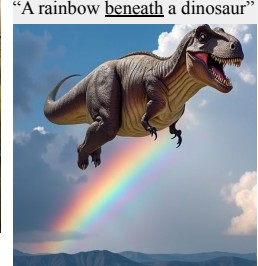

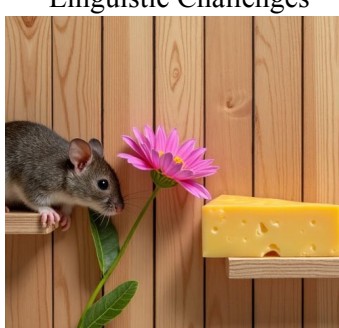

"A photo of a flower, a mouse, and cheese, but the mouse should not be above or below the flower and it should not be on the same side as the cheese."

### Many Positional Relations

Figure 2: Stitch excels at complex positional prompts.

et al., 2024). Despite these significant advances, positional control remains underexplored, and T2I models still struggle to manage object positions accurately (Bakr et al., 2023; Gokhale et al., 2022).

**Position control improvements.** Many base models improve spatial limitations during end-to-end training, for example with a joint understanding-generation objective (Wu et al., 2025b; Ma et al., 2025; Chen et al., 2025a;b; Deng et al., 2025) or prompt embeddings from Multimodal LLM (MLLM)-based encoders (Wu et al., 2025a; Fang et al., 2025; Wu et al., 2025c;d). In earlier T2I models, spatial guidance in forms such as bounding boxes could be used to improve positional control (Fang et al., 2025; Feng et al., 2023; Yang et al., 2024; Lian et al., 2024; Zhang et al., 2023b; 2024) often via generating and stitching sub-images. In fact, stitching images dates back to classical methods (Davis, 1998; Efros & Freeman, 2001), though modern approaches use deep networks. Methodologically, these externally position-guided modifications are related to tasks involving layout-conditioned T2I synthesis (Zhang et al., 2023a; Li et al., 2023; Farshad et al., 2023; Chen et al., 2024b; Mou et al., 2024; Ohanyan et al., 2024; Xie et al., 2023; Mo et al., 2024), which are more constrained by fine-grained user input. Unfortunately, these methods are implemented on older U-Net or Diffusion Transformer (Peebles & Xie, 2023) architectures. In contrast, many recent models adopt MMDiT (Esser et al., 2024) or related architectures (Wu et al., 2025a; BlackForest, 2024; Cai et al., 2025), where such techniques often fail to generalize effectively (Jiao et al., 2025). Instead, it has been shown that MMDiT-based architectures may be more amenable to generation-level edits like prediction swapping (Jiao et al., 2025; Bader et al., 2025), offering insights into MMDiT's inference behavior that we build upon. Concurrent works RAG (Chen et al., 2025c) and Regional Prompting (Chen et al., 2024a) also improve FLUX's position control by merging bounding box-generated objects. They mix predictions for full and sub-prompts within regions and over multiple steps, while Stitch composes regions once without mixing. Our Cutout enables tight foreground extraction from the regions, whereas these methods are limited to merging full regions.

**Position control benchmarks.** Many T2I benchmarks evaluate image generation capabilities, with GenEval (Ghosh et al., 2023) and T2I-CompBench++ (Huang et al., 2023; 2025) among the most widely used. Both include basic spatial reasoning categories of form $\langle obj_1\rangle\langle relation_{12}\rangle\langle obj_2\rangle$. While some benchmarks target positional understanding, many focus on basic formats lacking task complexity (Gokhale et al., 2022), or shift to related spatial tasks like shape generation (Sim et al., 2024) or 3D spatial reasoning (Wang et al., 2025b). Other general benchmarks include 2D spatial tasks (Li et al., 2025; Wang et al., 2025a) which may extend to 3 or 4 objects (Bakr et al., 2023; Feng et al., 2023), or evaluate prompts that are *natural* (Feng et al., 2023) or long (Hu et al., 2024), but lack explicit structure or precise positional tasks. Our proposed PosEval advances positional evaluation with a comprehensive, targeted suite of complex tasks.

## 3 METHODS

Despite progress in many aspects of image synthesis, position-related prompts continue to pose challenges for T2I generation models. To address this, we introduce Stitch, a flexible method to improve positional generation in models with MMDiT (Esser et al., 2024) or similar architectures.

### 3.1 PRELIMINARIES

Recent generative models define generation as a time-dependent transformation from Gaussian noise $x_0 \sim \mathcal{N}(0, I)$ to data $x_1 \sim p_{\text{data}}$, expressed as $x_\tau = \alpha_\tau x_0 + \sigma_\tau x_1$, with $\alpha_\tau$ decreasing and $\sigma_\tau$ increasing over time $\tau$. FM models (Albergo & Vanden-Eijnden, 2023; Liu et al., 2023a; Lipman et al., 2023; Esser et al., 2024) treat this trajectory as a differential equation, with learned approximation of its conditional vector field. Simulating this equation from noise to data in $T$ steps yields samples.

Many models operate in a VAE latent space (Kingma & Welling, 2022), encoding an image into latent token sequence $\mathcal{Z}_v = \{z_v^i\}_{i=1}^{N_v}$. To condition generation, a text encoder (e.g. T5 (Raffel et al., 2019) or CLIP (Radford et al., 2021)) embeds the prompt as $\mathcal{Z}_t = \{z_t^i\}_{i=1}^{N_t}$. Learnable projections $E_v$ and $E_t$ map both sets into a shared embedding space, yielding sets of *visual and text vectors*:

$$\mathcal{X}_v = \{x_v^i\}_{i=1}^{N_v} = E_v(\mathcal{Z}_v), \quad \mathcal{X}_t = \{x_t^i\}_{i=1}^{N_t} = E_t(\mathcal{Z}_t),$$

where each $x_v^i, x_t^i \in \mathbb{R}^d$ is a single vector in the corresponding sequence. MMDiT-like (Esser et al., 2024) architectures process $\mathcal{X}_v$ and $\mathcal{X}_t$ iteratively with a stack of time-conditioned transformer blocks. Each block first applies a pre-attention transformation $(\tilde{\mathcal{X}}_v, \tilde{\mathcal{X}}_t) = F_{\text{pre}}(\mathcal{X}_v, \mathcal{X}_t; \tau)$, which includes normalization and timestep modulation. Let $\tilde{\mathcal{X}} = \tilde{\mathcal{X}}_v \cup \tilde{\mathcal{X}}_t = \{\tilde{x}_i\}_{i=1}^N, N = N_v + N_t$ denote all pre-attention vectors. Each $\tilde{x}_i \in \tilde{\mathcal{X}}$ is mapped to queries, keys, and values via functions $Q, K, V$ derived from linear projections and normalization. Attention outputs are then computed as:

$$z(\tilde{x}_i) = \sum_{\tilde{x}_j \in \tilde{\mathcal{X}}} \text{softmax}_j\left(\frac{\langle Q(\tilde{x}_i), K(\tilde{x}_j)\rangle}{\sqrt{d}} + M(\tilde{x}_i, \tilde{x}_j)\right) V(\tilde{x}_j),$$

where $M : \tilde{\mathcal{X}} \times \tilde{\mathcal{X}} \to \{0, -\infty\}$ is an attention mask, and $\mathcal{Z} = \{z(\tilde{x}_i)\}_{i=1}^N$ denotes the set of attention outputs. Finally, a post-attention transformation $(\mathcal{X}_v^*, \mathcal{X}_t^*) = F_{\text{post}}(\mathcal{Z}, \mathcal{X}_v, \mathcal{X}_t; \tau)$ updates the representations. Attention is often multi-headed, where each head $h$ computes attention independently over $\tilde{\mathcal{X}}$ with its own $Q_h, K_h, V_h$, and the outputs are combined into $\mathcal{Z}$. Stitch builds upon MMDiT's design, particularly with targeted masking within the multimodal transformer blocks.

### 3.2 STITCH: TRAINING-FREE POSITION CONTROL FOR MMDIT

As seen in Figure 3, Stitch begins by using MLLM $L$ to decompose the full text prompt $P$ into $K$ sub-prompts $\{p_k\}_{k=1}^K$, each associated with a corresponding bounding box $\{b_k\}_{k=1}^K$, where $b_k = [x_{\min}, x_{\max}, y_{\min}, y_{\max}] \in \{0, \dots, W-1\}^4$. Each pair $(p_k, b_k)$ represents a distinct object in the scene. Additionally, $L$ generates a background prompt $p_0$, with a bounding box $b_0$ that spans the entire image. For the first $S$ steps, the FM model $F$ separately generates everything within the designated bounding boxes via Region Binding (detailed below). Once objects are adequately formed, their corresponding foreground latent tokens $\mathcal{Z}_{v,k}^S$ are cut from the resulting latent maps. We do so with our Cutout (discussed later in this section), the necessary information for which is obtained from a model-specific attention head. The extracted foreground latent tokens $\mathcal{Z}_{v,k}^S$ are combined with the common background latent tokens $\mathcal{Z}_{v,0}^S$ into a single composite latent $\mathcal{C}$, used exclusively for the remainder of the generation. The model proceeds without constraints and conditioned on full prompt $P$, allowing $F$ to enhance the overall quality and consistency while completing the image.

**Region Binding** To ensure objects are fully generated within their specified bounding boxes and to prevent capturing of partial fragments during foreground extraction, we introduce three attention-masking constraints. These constraints guide the model to focus generation solely within each bounding box, effectively isolating the object from the surrounding context.

Formally, for each sub-prompt $p_k$ and bounding box $b_k$, let $\tilde{\mathcal{X}}_{v,b_k} \subseteq \tilde{\mathcal{X}}_v$ denote the visual vectors inside the bounding box, $\tilde{\mathcal{X}}'_{v,b_k} = \tilde{\mathcal{X}}_v \setminus \tilde{\mathcal{X}}_{v,b_k}$ those outside, and $\tilde{\mathcal{X}}_{t,p_k}$ the text vectors corresponding

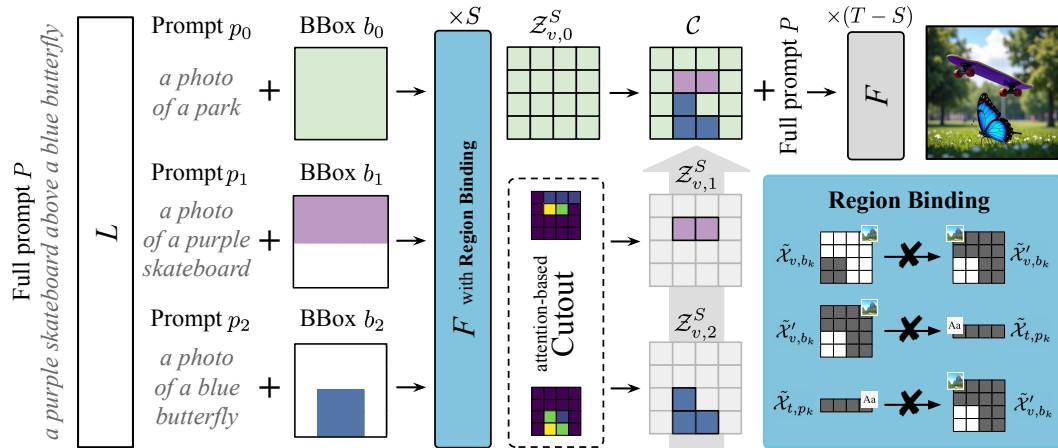

Figure 3: Stitch: Multimodal LLM $L$ splits full prompt $P$ into object prompts $p_k$ and bounding boxes $b_k$, along with full-image background prompt $p_0$. MMDiT-based model $F$ seperately sketches objects and background (*butterfly* ■, *skateboard* ■, *park* ■) for $S$ timesteps. With **Region Binding** attention-masking constraints, $F$ generates each $p_k$ in $b_k$. In **Cutout**, the highest attention weights in a targeted head select tighter latent regions linked to foreground objects $\mathcal{Z}_{v,k}^S$, which are merged with the background latents $\mathcal{Z}_{v,0}^S$ to form composite latent $\mathcal{C}$. For the remaining steps, the unconstrained $F$ seamlessly stitches the sketches into a coherent image conditioned on the full prompt.

to the sub-prompt. In every layer and head we apply the attention mask $M$ with the following constraints: (1) block attention from inside to outside the bounding box, (2) block attention from outside the bounding box to the text, (3) block attention from the text to outside the bounding box:

$$M(\tilde{\mathcal{X}}_{v,b_k}, \tilde{\mathcal{X}}'_{v,b_k}) = -\infty, \quad M(\tilde{\mathcal{X}}'_{v,b_k}, \tilde{\mathcal{X}}_{t,p_k}) = -\infty, \quad M(\tilde{\mathcal{X}}_{t,p_k}, \tilde{\mathcal{X}}'_{v,b_k}) = -\infty.$$

The background bounding box $b_0$ spans the entire image, so $\tilde{\mathcal{X}}_{v,b_0} = \tilde{\mathcal{X}}_v$, imposing no constraints.

**Cutout** To avoid visible seams from mismatched backgrounds, we extract the object latent tokens before constructing the composite latent $\mathcal{C}$. As generation is incomplete after step $S$, conventional segmentation tools like SAM (Kirillov et al., 2023) cannot be used to find the shape to cut out. However, we observe that in some attention heads, text vectors focus on foreground objects. Given such a head (selection discussed in Section 5.2), we construct the Cutout mask by selecting visual tokens in descending order of attention weight. The attention weight for each visual token is computed as the average attention that its corresponding visual vector gets from all non-padding text vectors. We select tokens until their attention weights sum to a fraction $\eta \in [0,1]$ of the total attention assigned to all visual tokens. The mask is then smoothed with 2D max pooling of kernel size $\kappa$.

## 4 POSEVAL: BENCHMARKING POSITIONAL GENERATION CAPABILITIES

As models improve on basic *Position* tasks of form $\langle obj_1 \rangle \langle relation_{12} \rangle \langle obj_2 \rangle$, broader benchmarks are needed to more thoroughly evaluate T2I spatial capabilities. We introduce PosEval, featuring tasks going beyond basic *Position* in difficulty, yet maintaining clear and specific task definitions. Each task is designed to isolate and evaluate a specific aspect of T2I capabilities. To ensure ease of use, PosEval builds upon GenEval, using Mask2Former-based (Cheng et al., 2022) object detection and procedural relation verification extended to the new tasks. We reuse the same set of objects $\langle obj \rangle$ and relations $\langle relation \rangle$, and adopt their protocol of generating 100 prompts per category and four images per prompt. Figure 4 and the Appendix show example prompts. The new tasks include:

**Two Objects (2 Obj):** For completeness, we inherit GenEval's *Position* task, unchanged.

**Three Objects (3 Obj):** We extend the basic *Position* task to three objects linked by two spatial relations. Objects $\langle obj_1 \rangle, \langle obj_2 \rangle, \langle obj_3 \rangle$ are arranged consecutively in a chain wrapped to fit on a $2 \times 2$ grid, ensuring one horizontal and one vertical relation. For a single $\langle relation_{ij} \rangle$ between

adjacent $\langle obj_i \rangle$ and $\langle obj_j \rangle$, we use $f(\langle relation_{ij} \rangle) :=$ "The $\langle obj_i \rangle$ is $\langle relation_{ij} \rangle$ the $\langle obj_j \rangle$" or "The $\langle obj_j \rangle$ is $\langle relation_{ji} \rangle$ the $\langle obj_i \rangle$", with equal probability. The task prompts are built in the form: "A photo of a $\langle obj_1 \rangle$, a $\langle obj_2 \rangle$, and a $\langle obj_3 \rangle$. $f(\langle relation_{12} \rangle)$. $f(\langle relation_{23} \rangle)$." Afterwards, objects in the first sentence are shuffled, along with sentences 2 and 3. Objects are listed upfront to avoid ambiguity. The model is tested on correctly interpreting the relations, not the underlying grid.

**Four Objects (4 Obj):** We further extend to four objects connected by four spatial relations, otherwise using the same setup as in the *3 Obj* task. The task prompts are built in the form: "A photo of a $\langle obj_1 \rangle$, a $\langle obj_2 \rangle$, a $\langle obj_3 \rangle$, and a $\langle obj_4 \rangle$. $f(\langle relation_{12} \rangle)$. $f(\langle relation_{23} \rangle)$. $f(\langle relation_{34} \rangle)$. $f(\langle relation_{41} \rangle)$." Again, objects in sentence 1 and sentences 2–5 are randomly permuted, and the model is evaluated on correctly interpreting the specified spatial relations.

**Positional Attribute Binding (PAB):** In the form "a $\langle attr_1 \rangle \langle obj_1 \rangle \langle relation_{12} \rangle$ a $\langle attr_2 \rangle \langle obj_2 \rangle$", PAB extends *Attribute Binding* ("a $\langle attr_1 \rangle \langle obj_1 \rangle$ and a $\langle attr_2 \rangle \langle obj_2 \rangle$") with inter-object positions.

**Negative Relations (Neg):** To evaluate understanding of *Negative Relations*, we use prompts of form, "a photo of a $\langle obj_1 \rangle$ and a $\langle obj_2 \rangle$, a $\langle obj_1 \rangle$ is not $\langle relation_{12} \rangle$ a $\langle obj_2 \rangle$", with objects listed for clarity. It is evaluated that: (1) both objects are present, and (2) $\langle obj_2 \rangle$ appears anywhere *except* in the specified relation to $\langle obj_1 \rangle$. Prompts are derived from GenEval's position prompts by replacing the relation with a negation of the opposite relation, keeping the same target image (e.g. "a photo of a dog right of a teddy bear" $\rightarrow$ "a photo of a dog and a teddy bear, a dog is not left of a teddy bear").

**Relative Relations (Rel):** We evaluate understanding of relations *relative to other relations*. Prompts contain three objects and two spatial relations, the first in the form $\langle relation_{ij} \rangle$ and the second defined relative to the first. Relative relations $\langle rel.relation \rangle$ can be same or opposite, each comprising half the prompts. The *same* relation is "on the same side of", while the *opposite* relations include "on the other side of", "on the opposite side of", and "on the contrary side of", for diversity. Prompts take the form, "a photo of a $\langle obj_1 \rangle$ $\langle relation_{12} \rangle$ $\langle obj_2 \rangle$, and a $\langle obj_3 \rangle$ $\langle rel.relation \rangle$ the $\langle obj_2 \rangle$ $\langle prep \rangle$ the $\langle obj_1 \rangle$", with $\langle prep \rangle \in \{$"for", "as"$\}$ chosen to be grammatically correct.

## 5 Main Experiments

We validate our three primary contributions: Stitch, Cutout, and PosEval. In Stitch, $S = 10$ for FLUX and SD3.5 and 6 for Qwen-Image. For all three models, we use $T = 50$, $\kappa = 5$, and bounding boxes $b_k$ are generated on a $W \times W = 32 \times 32$ grid with prompts $p_k$ from GPT-5 (OpenAI, 2025) and GPT-4 (OpenAI, 2023). $\eta$ is set to 0.95 for SD3.5 and FLUX. Results are on 3 seeds. All other parameters are the same as the base models. Our code is attached in the Supplementary Materials.

### 5.1 Enhancing Positional Understanding with Stitch

PosEval highlights that while SOTA models handle basic *Position* prompts well, they struggle with complexity, suggesting that the overall positional problem remains unsolved. Table 1 presents PosEval results, covering our five newly introduced tasks and GenEval's *Position* (*2 Obj*). While models like BLIP3-o (Chen et al., 2025a) and JanusPro (Chen et al., 2025b) perform the basic task well (87% and 79%, respectively) their accuracy drops sharply on harder tasks. On *Relative Relations*, they score only 8% and 11%, and when scaling to *Four Objects*, their accuracy falls to 2% and 4%. PosEval reveals greater shortcomings in top models than the basic *Position* task suggests.

Stitch consistently enhances base models on PosEval tasks, with up to 48 percentage point improvement (218% relative increase) over FLUX on *2 Obj* and 37 percentage point increase on PosEval overall (206% relative increase). This boost is also seen on *Position* tasks in other benchmarks (T2ICompBench (Huang et al., 2023) and HRS-Bench (Bakr et al., 2023) in the Appendix). Stitch + Qwen-Image, ranks first on 5/6 PosEval tasks and ties with Stitch + FLUX on the last. It has the highest overall accuracy, with a 54% relative gain over the previous best, LMD (Lian et al., 2024).

For Qwen-Image, FLUX, and SD3.5, Stitch refines positional details without degrading visual quality, shown in Figure 4 on all PosEval categories (SD3.5 in Appendix). Even scaling to four objects, it can integrate without awkward or unnatural seams. The boost observed in Table 1 from applying Stitch can be partly attributed to reduced positional errors, such as incorrectly placing or omitting objects. These mistakes are seen when FLUX omits the oven in *Four Object*, and Qwen-Image

Table 1: PosEval reveals that leading T2I models still struggle with complex positional prompts. However, Stitch boosts positional generation on 3 different MMDiT-based models, achieving SOTA.

| Model | 2 Obj | 3 Obj | 4 Obj | Neg | Rel | PAB | Avg.↑ |
|---|---|---|---|---|---|---|---|
| *No layout guidance* | | | | | | | |
| FLUX.1 [Dev] (BlackForest, 2024) | 0.22 | $0.06_{\pm0.01}$ | $0.02_{\pm0.00}$ | $0.62_{\pm0.01}$ | $0.03_{\pm0.01}$ | $0.15_{\pm0.02}$ | 0.18 |
| SD3 Medium (Esser et al., 2024) | 0.34 | $0.05_{\pm0.01}$ | $0.01_{\pm0.00}$ | $0.62_{\pm0.02}$ | $0.05_{\pm0.01}$ | $0.14_{\pm0.01}$ | 0.20 |
| SD3.5 Large (Esser et al., 2024) | 0.34 | $0.06_{\pm0.00}$ | $0.02_{\pm0.01}$ | $0.64_{\pm0.03}$ | $0.06_{\pm0.02}$ | $0.16_{\pm0.01}$ | 0.21 |
| HiDream-I1-Full (Cai et al., 2025) | 0.60 | $0.19_{\pm0.01}$ | $0.10_{\pm0.00}$ | $0.66_{\pm0.00}$ | $0.09_{\pm0.01}$ | $0.29_{\pm0.01}$ | 0.32 |
| BAGEL (Deng et al., 2025) | 0.64 | $0.23_{\pm0.01}$ | $0.16_{\pm0.02}$ | $0.73_{\pm0.01}$ | $0.07_{\pm0.01}$ | $0.22_{\pm0.02}$ | 0.34 |
| OpenUni-B-512 (Wu et al., 2025d) | 0.77 | $0.09_{\pm0.01}$ | $0.03_{\pm0.00}$ | $0.56_{\pm0.01}$ | $0.01_{\pm0.00}$ | $0.60_{\pm0.01}$ | 0.34 |
| Janus-Pro (Chen et al., 2025b) | 0.79 | $0.14_{\pm0.01}$ | $0.04_{\pm0.01}$ | $0.69_{\pm0.02}$ | $0.11_{\pm0.00}$ | $0.46_{\pm0.03}$ | 0.37 |
| BLIP3-o (Chen et al., 2025a) | $\mathbf{0.87}_{\pm0.01}$ | $0.15_{\pm0.00}$ | $0.02_{\pm0.00}$ | $0.55_{\pm0.01}$ | $0.08_{\pm0.01}$ | $0.63_{\pm0.01}$ | 0.38 |
| Qwen-Image (Wu et al., 2025a) | 0.76 | $0.40_{\pm0.01}$ | $0.21_{\pm0.02}$ | $0.49_{\pm0.00}$ | $0.10_{\pm0.01}$ | $0.61_{\pm0.03}$ | 0.43 |
| *Layout guidance, with training* | | | | | | | |
| GoT (Fang et al., 2025) | 0.34 | $0.02_{\pm0.00}$ | $0.00_{\pm0.00}$ | $0.52_{\pm0.01}$ | $0.02_{\pm0.01}$ | $0.09_{\pm0.01}$ | 0.17 |
| LayoutGPT (Feng et al., 2023) | $0.41_{\pm0.04}$ | $0.18_{\pm0.03}$ | $0.10_{\pm0.02}$ | $0.37_{\pm0.03}$ | $0.25_{\pm0.02}$ | $0.07_{\pm0.02}$ | 0.23 |
| *Layout guidance, training-free* | | | | | | | |
| RPG (Yang et al., 2024) | $0.28_{\pm0.00}$ | $0.07_{\pm0.01}$ | $0.01_{\pm0.00}$ | $0.65_{\pm0.00}$ | $0.06_{\pm0.01}$ | $0.14_{\pm0.01}$ | 0.20 |
| RAG (Chen et al., 2025c) | $0.50_{\pm0.00}$ | $0.11_{\pm0.00}$ | $0.04_{\pm0.01}$ | $0.61_{\pm0.01}$ | $0.13_{\pm0.01}$ | $0.21_{\pm0.01}$ | 0.27 |
| Reg. Prompting (Chen et al., 2024a) | $0.25_{\pm0.01}$ | $0.17_{\pm0.00}$ | $0.28_{\pm0.03}$ | $0.68_{\pm0.00}$ | $0.30_{\pm0.01}$ | $0.19_{\pm0.01}$ | 0.31 |
| LMD (Lian et al., 2024) | $0.74_{\pm0.01}$ | $0.43_{\pm0.02}$ | $0.23_{\pm0.01}$ | $0.54_{\pm0.01}$ | $0.35_{\pm0.01}$ | $0.46_{\pm0.01}$ | 0.46 |
| **Stitch (ours) + SD3.5 Large** | $0.53_{\pm0.02}$ | $0.22_{\pm0.03}$ | $0.12_{\pm0.01}$ | $0.79_{\pm0.00}$ | $0.27_{\pm0.02}$ | $0.37_{\pm0.02}$ | 0.38 |
| Gain over SD3.5 Large | +0.19 | +0.16 | +0.10 | +0.15 | +0.21 | +0.21 | +0.17 |
| **Stitch (ours) + FLUX.1 [Dev]** | $0.70_{\pm0.02}$ | $0.44_{\pm0.01}$ | $0.38_{\pm0.01}$ | $0.83_{\pm0.02}$ | $\mathbf{0.48}_{\pm0.02}$ | $0.44_{\pm0.00}$ | 0.55 |
| Gain over FLUX.1 [Dev] | +0.48 | +0.38 | +0.36 | +0.21 | +0.45 | +0.29 | +0.37 |
| **Stitch (ours) + Qwen-Image** | $\mathbf{0.87}_{\pm0.01}$ | $\mathbf{0.67}_{\pm0.01}$ | $\mathbf{0.61}_{\pm0.01}$ | $\mathbf{0.93}_{\pm0.00}$ | $0.43_{\pm0.01}$ | $\mathbf{0.77}_{\pm0.00}$ | **0.71** |
| Gain over Qwen-Image | +0.11 | +0.27 | +0.40 | +0.44 | +0.33 | +0.16 | +0.28 |

incorrectly places the cow left of the laptop in *Negative*. Moreover, Stitch demonstrates strong comprehension of linguistically challenging prompts, like those in *Negative* and *Relative Relation*.

Figure 2 further highlights Stitch's handling of complex scenes on FLUX. On the left, we show how it successfully interprets a prompt that combines all six PosEval task categories. Despite the complexity (featuring four distinct objects, multiple attributes, and both negative and relative spatial relations), Stitch generates a coherent and semantically accurate image. In the center, we demonstrate that it enhances FLUX's ability to generate rare and challenging combinations, such as a dinosaur *above* a rainbow. On the right, we show that Stitch's strong language understanding enables it to interpret even confusing text and translate it into coherent visual arrangements.

## 5.2 Assessing Foreground Segmentation Accuracy of Cutout

We choose per-model attention heads by generating $80$ images with the prompt "a photo of a ⟨obj⟩" using the 80 GenEval objects. After step $S$, we save text-to-image attention averaged over non-padding tokens, avoiding unwanted behavior introduced by padding. We test thresholds $\eta \in \{0.75, 0.80, 0.85, 0.90, 0.95, 0.97, 0.99\}$ and extract SAM (Kirillov et al., 2023) masks generated from the final images to use for ground truth, excluding objects with inadequate SAM masks. We evaluate the Intersection over Union (IoU) between masks generated by each attention head across the different $\eta$ values, choosing the head-threshold combination with the highest IoU. With this procedure, a Cutout head can be selected once per model and applied at test time for all images.

In Appendix F, we present the top five attention heads for each base model, ranked by IoU with SAM maps and displayed with each head's optimal threshold $\mu$. We also report Intersection over Target (IoT), calculated as $\frac{|\text{Prediction} \cap \text{Target}|}{|\text{Target}|}$. For each model, several attention heads exhibit high IoU, as illustrated for FLUX in Figure 18 (Appendix K). Key for Cutout, IoT reflects how fully the segmentation captures the foreground. Together, these metrics highlight which heads are well-

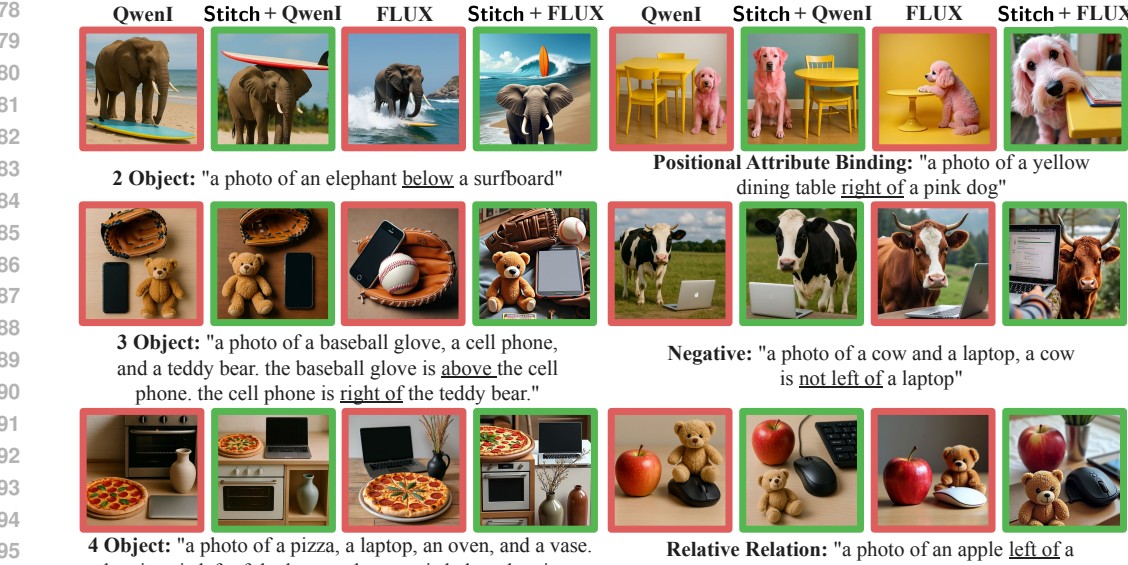

2 **Object:** "a photo of an elephant below a surfboard"

**Positional Attribute Binding:** "a photo of a yellow dining table right of a pink dog"

3 **Object:** "a photo of a baseball glove, a cell phone, and a teddy bear. the baseball glove is above the cell phone. the cell phone is right of the teddy bear."

**Negative:** "a photo of a cow and a laptop, a cow is not left of a laptop"

4 **Object:** "a photo of a pizza, a laptop, an oven, and a vase. the pizza is left of the laptop. the oven is below the pizza. the vase is right of the oven. the vase is below the laptop."

**Relative Relation:** "a photo of an apple left of a computer mouse, and a teddy bear on the same side of the computer mouse as the apple"

Figure 4: Stitch corrects Qwen-Image (QwenI) and FLUX position on PosEval without quality loss.

suited for Cutout; for example, FLUX block 14, head 20 achieves an IoU of 62% and an IoT of 92% (see Appendix F). Example masks are presented in Figure 6 for this head, selected for FLUX Cutout. We observe that Cutout masks often leave a narrow border around the object, perhaps because low-level details have not yet solidified, helping explain their better IoT compared to IoU.

This behavior suits Stitch's purpose: the border does not degrade the images and helps capture the full object, although even small missing parts can be reconstructed in later generation steps.

To assess Cutout head generalizability, we evaluate FLUX's Cutout head (Block 14, Head 20) on 80 single-object validation prompts, evenly split across eight categories (Appendix J). The resulting IoU (63%) and IoT (90%) closely match those from the GenEval-based prompts and remain highly consistent across categories, as shown in Figure 5 (and Table 9), with IoU exhibiting only a 0.06 standard deviation.

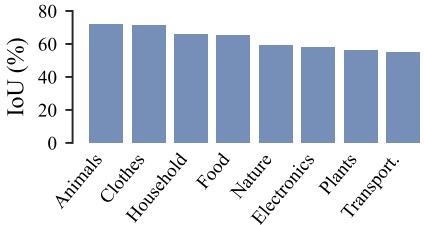

Figure 5: Cutout head generalizes across diverse validation categories.

### 5.3 ASSURING RELIABILITY OF OUR POSEVAL BENCHMARK

PosEval adopts the same evaluation protocol as GenEval (Ghosh et al., 2023), whose components were thoroughly validated by the original authors. We conduct a user study to validate the reliability of the evaluation platform for the newly introduced benchmark categories. For this study, we create a candidate image pool with 4 images per prompt from Qwen-Image, Stitch + Qwen-Image, FLUX, Stitch + FLUX, SD3.5, and Stitch + SD3.5. We divide this pool into images labeled as *correct* and *incorrect* by the automated evaluation, then randomly select 50 images from each group. Three annotators independently evaluate whether each image "correctly follows the prompt", and we report their level of agreement with the automatic evaluation. Figure 7 shows the evaluation protocol performs consistently (table in Appendix), with all PosEval categories within 10% of the average inter-annotator agreement.

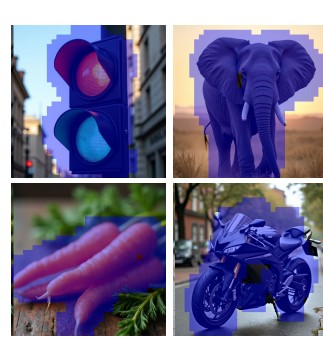

Figure 6: Cutout cleanly extracts objects mid-generation.

Each PosEval task is challenging yet complementary, allowing insights through comparison. Comparing *2 Obj*, *3 Obj*, and *4 Obj* reveals how models handle positional generation as object count and relationships grow. As shown in Table 1, strong *2 Obj* performance does not guarantee scaling. BLIP3-o (Chen et al., 2025a) scores $87\%$ on *2 Obj* but drops to $15\%$ and $2\%$ on *3 Obj* and *4 Obj*, while Qwen-Image (Wu et al., 2025a), with lower initial scores, maintains better performance at $40\%$ and $21\%$, respectively. This suggests some models may overfit to previously seen tasks without effectively handling positional generation as complexity grows. Moreover, although *Negative* prompts generate the same images as *2 Obj* prompts, they are easier to satisfy due to looser criteria

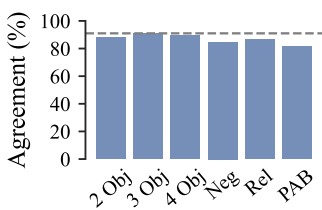

Figure 7: PosEval's evaluation is human-aligned, near average annotator agreement (line - -).

(e.g., "not right" vs. "left"). If a model scores higher on *2 Obj* than *Neg* (e.g. Janus-Pro's $79\%$ vs. $69\%$), the gap likely reflects prompt interpretation issues rather than image generation. Similarly, both *Rel* and *3 Obj* include three objects and two spatial relations, but *Rel* is linguistically more complex and restricts relations to one axis, while *3 Obj* spans two axes. For example, Qwen-Image scores $40\%$ on *3 Obj* but only $10\%$ on *Rel*, and $76\%$ on *2 Obj* versus $49\%$ on *Neg*, indicating difficulty with complex prompts.

## 6 ADDITIONAL ANALYSIS AND ABLATIONS OF STITCH

We investigate Stitch in greater detail. First, we analyze how masks generated with Cutout compare to SAM masks in terms of IoU and IoT as we vary $\eta$ (see Figure 19, Appendix L), focusing on the top 10 FLUX heads. Top heads stay best for IoU across all $\eta$, while IoT rises, allowing us to select the Cutout head once and tune $\eta$ to reduce missed object parts. Next, we evaluate Stitch + FLUX as $\eta$ is increased from 0.75 to 0.99 (see Table 11, Appendix L). We test accuracy on PosEval and use a human study to evaluate *Blend* on the basic GenEval *Position* task, *2 Obj*. We define an image having *Blend* as being "visually coherent (and not like many images put side by side)". Figure 8 illustrates this effect: the right image (with Cutout) exhibits *Blend*, while the left image (without Cutout) does not. Increasing $\eta$ boosts PosEval accuracy but can harm *Blend* if set too high.

Next, Table 2 presents an ablation study on FLUX to evaluate the contributions of Region Binding, Cutout, and raising $\eta$ to 0.95 to widen the object buffer. For each setup, three people independently and binarily decide if each of 100 images (one image per prompt) displays *Blend*, and we report the percentage of images for which they answer *yes*. As shown in Table 2, the Region Binding is the primary driver of Stitch's position-related performance improvements (e.g. boosting *2 Obj* from $22\%$ to $81\%$). However, combining full bounding boxes requires using backgrounds from the partially generated Region Binding predictions, which may contain conflict-

Without Cutout        With Cutout

"A photo of a laptop left of a cow."

Figure 8: Cutout boosts coherence.

ing color information; this can hinder the model's ability to seamlessly stitch components into a coherent image. Consequently, Region Binding alone fails to properly blend $32\%$ of images. Adding Cutout improves the *Blend* success rate to $99\%$ of images, though at the cost of some positional accuracy. Raising the Cutout threshold to $\eta = 0.95$ recovers lost performance. While lower thresholds often aid in selecting heads focused on the foreground, a higher threshold is preferable afterward, as Stitch benefits from higher IoT. As shown in Table 2, this improves accuracy without significantly affecting *Blend*. This final combination is what allows Stitch to achieve both precise positional control and high image quality.

Since Stitch relies on MLLM-generated bounding boxes, we assess its sensitivity to them (Table 12, Appendix O) using two strategies: (1) perturbing the original boxes by shifting their edges up to $10\%$ while preserving the overall layout, and (2) shuffling the sub-prompts within the boxes, which can disrupt the layout. Our results show that Stitch is robust to small positional shifts but remains sensitive to changes in the overall layout accuracy.

We investigate timestep selection $S$ for Region Binding on FLUX. In Figure 9, we vary the number of timesteps $S$ on the basic *Position* task and compare to *Blend*. Selecting later steps improves *Po-*

Table 2: While Region Binding (RB) drives Stitch's positional control, Cutout improves the blend. By increasing Cutout threshold $\eta$ to 0.95, we recover accuracy without sacrificing blend.

| RB | Cutout | $\eta = 0.95$ | 2 Obj | 3 Obj | 4 Obj | Neg | Rel | PAB | Blend |
|----|--------|---------------|-------|-------|-------|-----|-----|-----|-------|
| | | | 0.22 | $0.06_{\pm0.01}$ | $0.02_{\pm0.00}$ | $0.62_{\pm0.01}$ | $0.03_{\pm0.01}$ | $0.15_{\pm0.02}$ | 1.0 |
| ✓ | | | $0.81_{\pm0.02}$ | $0.58_{\pm0.03}$ | $0.52_{\pm0.02}$ | $0.90_{\pm0.01}$ | $0.63_{\pm0.01}$ | $0.51_{\pm0.01}$ | 0.68 |
| ✓ | ✓ | | $0.51_{\pm0.01}$ | $0.30_{\pm0.01}$ | $0.18_{\pm0.01}$ | $0.71_{\pm0.03}$ | $0.31_{\pm0.04}$ | $0.29_{\pm0.01}$ | 0.99 |
| ✓ | ✓ | ✓ | $0.70_{\pm0.02}$ | $0.44_{\pm0.01}$ | $0.38_{\pm0.01}$ | $0.83_{\pm0.02}$ | $0.48_{\pm0.02}$ | $0.44_{\pm0.00}$ | 0.95 |

*sition* accuracy but reduces *Blend*, with a steep decline occurring between 10 and 20 steps. Offering the best balance, we select timestep 10 for FLUX. Similarly, we choose step 10 for SD3.5 and 6 for Qwen-Image.

Preserving base model quality while enhancing positional capabilities is essential. To validate, we compare each base model's quality with Stitch's via Aesthetic Score (Schuhmann & Beaumont, 2022). Across four images per prompt on the six PosEval tasks, Stitch results in only marginal changes: from $6.2 \pm 0.8$ to $6.1 \pm 0.8$ with Qwen, $6.3 \pm 0.8$ to $6.1 \pm 0.7$ with FLUX, and $5.3 \pm 0.8$ to $5.1 \pm 0.7$ with SD3.5. This indicates that Stitch does not significantly degrade image quality.

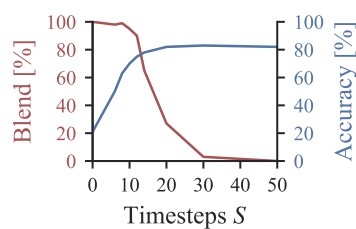

To ensure diversity is preserved, we compute per-prompt mean pairwise distance between samples in DINOv2 (Oquab et al., 2024) embedding space. We find average diversity is preserved or improved: FLUX increases from $0.34 \pm 0.15$ to $0.38 \pm 0.16$; SD3.5 remains at $0.43 \pm 0.15$ from $0.43 \pm 0.14$; and Qwen-Image rises from $0.16 \pm 0.11$ to $0.22 \pm 0.10$.

Figure 9: In Stitch, increasing the Region Binding timesteps $S$ improves accuracy but reduces blend.

## 7 CONCLUSION

We introduce Stitch, a training-free method for enhancing position-related T2I generation in MMDiT-based models. Stitch generates coherent, position-accurate images by first generating individual objects constrained to LLM-generated bounding boxes with our novel Region Binding constraints. With our Cutout, it then employs a targeted attention head to extract and combine these objects, before completing the image with the full prompt, unconstrained. We evaluate Stitch on PosEval, our new benchmark extending GenEval with five challenging, position-focused tasks. Evaluating top models on PosEval reveals that even the strongest T2I models still struggle with position-related tasks, despite performing well on basic *Position* tasks. Yet Stitch significantly boosts Qwen-Image, FLUX, SD3.5, achieving SOTA results while remaining entirely training-free.

## 8 REPRODUCIBILITY STATEMENT

We ensure the reproducibility of our experiments by providing detailed descriptions in Sections 5, 6, and the Appendix. The Supplementary Materials contain the implementation code, the prompts used in our benchmark, the bounding boxes generated by our method, as well as a README.md file with detailed reproduction instructions.

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

CONTENTS OF APPENDIX

## A  USE OF LLMS

This work used LLMs to improve sentence clarity and flow.

## B  STITCH RESULTS ON EXISTING BENCHMARKS

Table 3: Results on tasks from existing benchmarks

| Model | GenEval Position | T2ICompBench Spatial | HRS-Bench (spatial) Easy | Medium | Hard |
|---|---|---|---|---|---|
| **SD3.5 Large** | 0.34 | 0.22 | 0.46 | 0.19 | 0.08 |
| **Stitch (ours) + SD3.5 Large** | 0.53 | 0.28 | 0.54 | 0.26 | 0.08 |
| Gain over SD3.5 Large | +0.19 | +0.06 | +0.08 | +0.07 | +0.00 |
| **FLUX.1 [Dev]** | 0.22 | 0.25 | 0.50 | 0.23 | 0.08 |
| **Stitch (ours) + FLUX.1 [Dev]** | 0.70 | 0.44 | 0.82 | 0.52 | 0.23 |
| Gain over FLUX.1 [Dev] | +0.48 | +0.19 | +0.32 | +0.29 | +0.15 |
| **Qwen-Image** | 0.76 | 0.36 | 0.73 | 0.45 | 0.25 |
| **Stitch (ours) + Qwen-Image** | 0.85 | 0.50 | 0.88 | 0.63 | 0.40 |
| Gain over Qwen-Image | +0.09 | +0.14 | +0.15 | +0.18 | +0.15 |

We present results on several *Position*-related tasks in existing benchmarks: GenEval's *Position* task (Ghosh et al., 2023), T2I CompBench's *Spatial* task (Huang et al., 2023) (referred to as *2D-Spatial* in T2ICompBench++ (Huang et al., 2025)), and HRS-Bench's *Spatial* task (Bakr et al., 2023). T2I CompBench's *Spatial* task includes two objects and a single relation. HRS Bench's *Spatial* task is divided into three sub-categories: *easy* prompts include two objects and one relation; *medium* prompts include three objects and one or two relations; and *hard* prompts include four objects and one or two relations.

With Stitch, we improve all three base models across all tasks, except HRS-Bench *Hard* with SD3.5, where performance remains stable. This indicates that Stitch consistently enhances positional information across *Position*-related tasks from multiple benchmarks.

## C  POSEVAL USER STUDY RESULTS

We include the numerical results of our user study verifying PosEval's automated evaluation in Table 4. As mentioned in the main paper, all results are within 10% of the inter-annotator alignment (91%).

Table 4: PosEval alignment with human annotators. Average inter-annotator alignment is 91%.

| 2 Obj | 3 Obj | 4 Obj | Neg | Rel | PAB |
|---|---|---|---|---|---|
| 87.7% | 90.7% | 89.7% | 84.7% | 86.3% | 81.3% |

# D    ADDITIONAL QUALITATIVE EXAMPLES

Figure 10 presents qualitative examples for Stitch combined with SD3.5, showcasing two examples from each PosEval benchmark task. Additional qualitative results are provided for Stitch with FLUX in Figure 12, and for Stitch with Qwen-Image in Figure 11, also on two prompts per task.

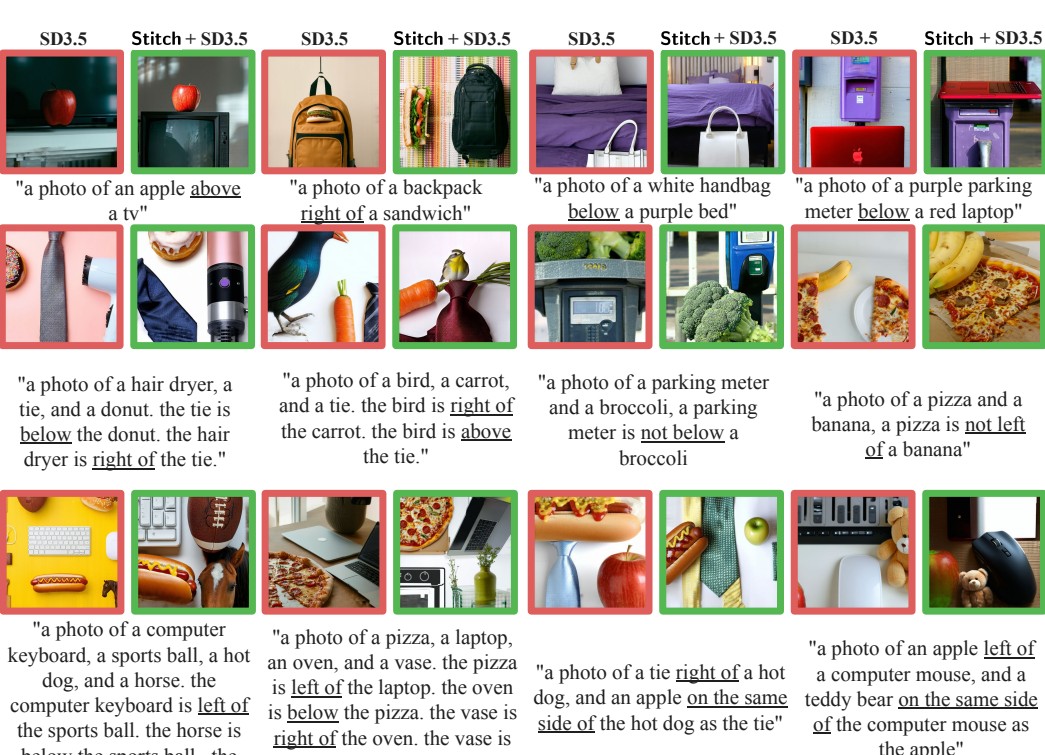

Figure 10: Qualitative examples for Stitch + SD3.5.

| QwenI | Stitch + QwenI | QwenI | Stitch + QwenI | QwenI | Stitch + QwenI | QwenI | Stitch + QwenI |
|---|---|---|---|---|---|---|---|

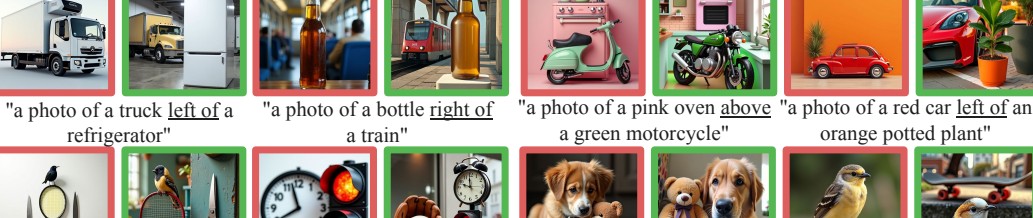

"a photo of a computer mouse left of a bench"

"a photo of a bird left of a couch"

"a photo of an orange snowboard above a green cat"

"a photo of a pink skateboard above a black train"

"a photo of a cell phone, a traffic light, and a skis. the traffic light is right of the skis. the cell phone is above the traffic light."

"a photo of a toilet, a laptop, and a tennis racket. the toilet is right of the laptop. the laptop is below the tennis racket."

"a photo of a tie and a baseball bat, a tie is not left of a baseball bat

"a photo of a wine glass and a kite, a wine glass is not below a kite"

"a photo of a toilet, a cat, a cell phone, and a frisbee. the toilet is above the cell phone. the frisbee is right of the cell phone. the cat is above the frisbee. the toilet is left of the cat."

"a photo of an apple, a spoon, a cake, and a toaster. the spoon is right of the apple. the cake is left of the toaster. the cake is below the apple. the spoon is above the toaster."

"a photo of a toilet left of a motorcycle, and a boat on the same side of the motorcycle as the toilet"

"a photo of a spoon right of a donut, and a computer keyboard on the same side of the donut as the spoon"

Figure 11: Additional qualitative examples for Stitch + Qwen-Image (QwenI).

| FLUX | Stitch + FLUX | FLUX | Stitch + FLUX | FLUX | Stitch + FLUX | FLUX | Stitch + FLUX |
|---|---|---|---|---|---|---|---|

"a photo of a truck left of a refrigerator"

"a photo of a bottle right of a train"

"a photo of a pink oven above a green motorcycle"

"a photo of a red car left of an orange potted plant"

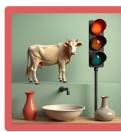
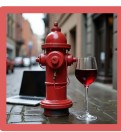
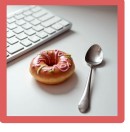

"a photo of a scissors, a tennis racket, and a bird. the tennis racket is below the bird. the scissors is right of the tennis racket."

"a photo of a clock, a baseball glove, and a traffic light. the baseball glove is left of the traffic light. the clock is above the traffic light."

"a photo of a dog and a teddy bear, a dog is not left of a teddy bear"

"a photo of a bird and a skateboard, a bird is not above a skateboard"

"a photo of a traffic light, a cow, a vase, and a sink. the cow is above the sink. the cow is right of the traffic light. the vase is left of the sink. the base is below the traffic light."

"a photo of a toilet, a donut, a sports ball, and a fire hydrant. the donut is above the fire hydrant. the sports ball is left of the fire hydrant. the sports ball is below the toilet. the toilet is left of the donut."

"a photo of a wine glass right of a fire hydrant, and a laptop on the same side of the fire hydrant as the wine glass"

"a photo of a spoon right of a donut, and a computer keyboard on the same side of the donut as the spoon"

Figure 12: Additional qualitative examples for Stitch + FLUX.

# E    COMPARISON OF POSEVAL WITH OTHER BENCHMARKS

We include an explicit comparison between PosEval and previously existing benchmark tasks which include position-based information in Table 5. We highlight several key aspects, focused on what is included in the benchmark and how specifically it is evaluated. First, we look at the number of objects present in positional tasks, starting with if they distinguish by the number of objects (*Dist. # Obj.*). For some benchmarks, such as the NSR-1k natural task (Feng et al., 2023), benchmark settings are not explicit about the number of objects present in each image. While all included benchmarks include 2-object prompts, we also specify if they include 3-object (*3 Obj.*) and 4-object (*4 Obj.*) prompts. Similar to *Dist. # Obj.*, we indicate if the tasks are generally focused (*Focused tasks*), meaning that they include settings which separate the positional problem into specific sub-tasks. An example of non-focused tasks is DPG (Hu et al., 2024) which targets the understanding of long prompts, but as a result does not sub-divide if prompts also test attribute binding, what types of positions are used (e.g. 2D vs. 3D), etc. In *Attr.*, we indicate if attributes (e.g. color) are present. Finally, *Neg. Pos.* shows the presence of *negative* in the sense of *position* (e.g. being "not left").

First, we observe that no existing benchmark incorporates the concept of *negative position*. Additionally, none of the prior benchmarks simultaneously include both attributes and focused tasks. However, focused tasks are essential for thoroughly evaluating and advancing positional generation. They enable developers to pinpoint where models fail and clearly compare capabilities. This is a key advantage of our PosEval, which addresses a gap left by previous benchmarks. In particular, our *Positional Attribute Binding* task introduces a uniquely challenging and previously unexplored dimension. Our *Relative Relation* task is also the first to specifically explore these types of more challenging *relative* relations, in an isolated and focused way. To ensure consistency with prior work, we also scale the number of objects to four. Overall, PosEval fills a critical gap in the literature by offering focused positional tasks while appropriately scaling the difficulty.

Table 5: PosEval comparison with other existing benchmarks

| | Dist. # Obj. | 3 Obj. | 4 Obj. | Focused tasks | Attr. | Neg. Pos. |
|---|---|---|---|---|---|---|
| HRS-Bench (Bakr et al., 2023) | ✓ | ✓ | ✓ | ✓ | ✗ | ✗ |
| NSR-1K - template (Feng et al., 2023) | ✓ | ✗ | ✗ | ✓ | ✗ | ✗ |
| NSR-1K - natural (Feng et al., 2023) | ✗ | ✓ | ✓ | ✗ | ✓ | ✗ |
| VISOR (Gokhale et al., 2022) | ✓ | ✗ | ✗ | ✓ | ✗ | ✗ |
| DPG (Hu et al., 2024) | ✗ | ✓ | ✓ | ✗ | ✓ | ✗ |
| T2I CompBench++ (Huang et al., 2025) | ✓ | ✗ | ✗ | ✓ | ✗ | ✗ |
| GenEval (Ghosh et al., 2023) | ✓ | ✗ | ✗ | ✓ | ✗ | ✗ |
| **PosEval (ours)** | ✓ | ✓ | ✓ | ✓ | ✓ | ✓ |

# F    TOP CUTOUT SEGMENTATION HEADS FOR EACH MODEL

We present the results of the top attention heads for FLUX (Table 6), Qwen-Image (Table 8), and SD3.5 (Table 7). The bold results are those selected for Cutout for each model. This is block 14 head 20 for FLUX, block 14 head 34 for SD3.5, and block 25 head 1 for Qwen-Image. Interestingly, we find that the best segmentation heads typically fall in the middle blocks (between 10 and 20 for FLUX and SD3.5, and between 15 and 30 for Qwen-Image, which has more blocks).

We include additional visualizations of the Cutout masks in FLUX in Figure 14. For six example objects, we present: (1) the final image, (2) the attention map from the Cutout head (block 14, head 20 in FLUX) at step $S = 10$, and (3) the final Cutout masks thresholded at $95\%$. By comparing the attention maps to the final images, we observe that the selected head already encodes information about the object's location in the latent space, even in the early stages of generation. Furthermore, the Cutout masks accurately capture the object's position, with a slight margin from the $95\%$ threshold. This buffer proves beneficial to Stitch, as confirmed by our ablation study.

In Figure 13, we also include segmentation maps for SD3.5, similar to those provided for FLUX in the main paper. These are compared to the SAM maps, for reference.

Table 6: Best segmentation heads for FLUX.

| Block | Head | $\eta$ | IoU | IoT |
|-------|------|--------|-----|-----|
| **14** | **20** | 0.75 | **0.62** | **0.92** |
| 17 | 23 | 0.75 | 0.56 | 0.82 |
| 13 | 21 | 0.75 | 0.54 | 0.87 |
| 14 | 15 | 0.75 | 0.54 | 0.87 |
| 10 | 1 | 0.75 | 0.53 | 0.86 |

Table 7: Best segmentation heads for SD3.5.

| Block | Head | $\eta$ | IOU | IOT |
|-------|------|--------|-----|-----|
| **14** | **34** | 0.75 | **0.56** | **0.94** |
| 17 | 13 | 0.75 | 0.56 | 0.96 |
| 15 | 31 | 0.75 | 0.55 | 0.94 |
| 18 | 36 | 0.75 | 0.55 | 0.96 |
| 14 | 6 | 0.80 | 0.55 | 0.90 |

Table 8: Best segmentation heads for Qwen-Image.

| Block | Head | $\eta$ | IOU | IOT |
|-------|------|--------|-----|-----|
| **25** | **1** | **0.9** | **0.55** | **0.86** |
| 27 | 17 | 0.97 | 0.53 | 0.89 |
| 26 | 4 | 0.85 | 0.53 | 0.82 |
| 19 | 12 | 0.97 | 0.53 | 0.90 |
| 18 | 11 | 0.85 | 0.52 | 0.84 |

SAM      Cutout

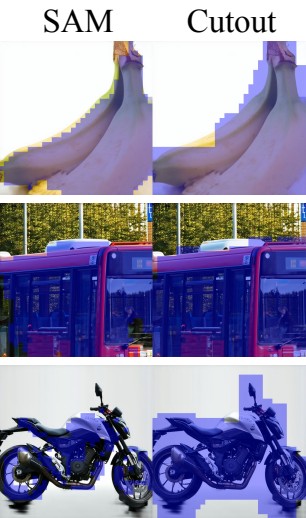

Figure 13: Segmentation head maps for SD3.5.

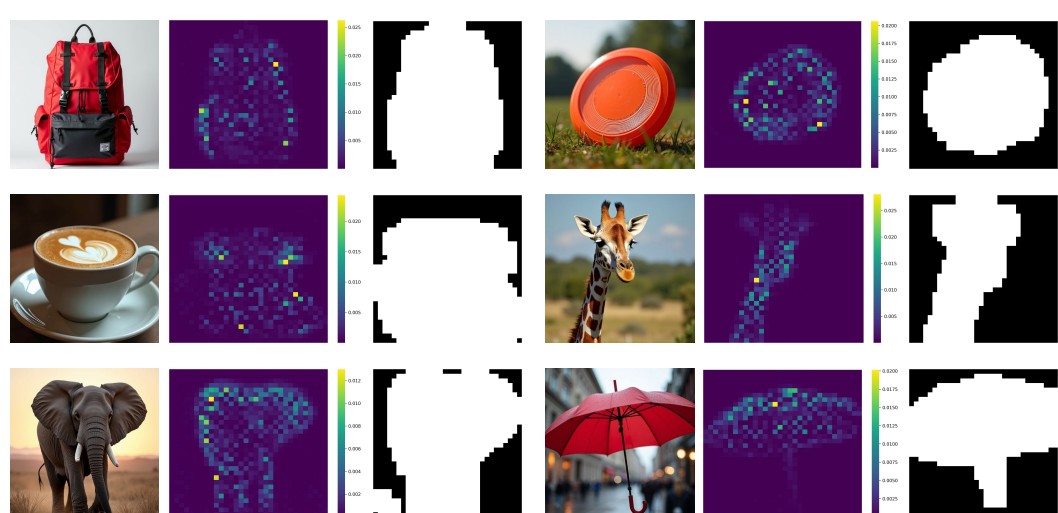

Figure 14: Attention weights and Cutout masks from head 20 in block 14 of FLUX, at step 10 of 50.

## G  BOUNDING BOXES AND SUB-PROMPTS GENERATION

To generate object bounding boxes $b_k$ and prompts $p_k$, $\forall k > 0$, we utilize GPT-5 (OpenAI, 2025) model with reasoning effort set to minimal and system prompt:

```
You have a canvas of size {W}x{W}. Decompose the given description
into single objects. Do not merge multiple objects into one box.
Fully cover the canvas with the object boxes. Avoid overlaps.
Do not leave space for background. Write 3 sentence justification
and then output just valid JSON in the format:
[
    {"prompt": "<object with properties only mentioned in the
    description>", "x_min": , "y_min": , "x_max": , "y_max": },
    ...
]
```

Each prompt $P$ is processed with user prompt:

```
Description: {P}
```

To generate background prompt $p_0$ we use GPT-4 (OpenAI, 2023) with system prompt:

```
Provide a simple fitting background description for this scene.
The background must not mention any of the specific objects or
elements from the prompt. The background must not mention any
other specific objects or people. Return only ONE word of the
background text, nothing else.
```

Given a prompt $P$ the user prompt again takes form:

```
Description: {P}
```

## H    USER STUDY INTERFACE

In Figures 15 and 16, we include and example of the user interface for our user studies to measure PosEval's quality and *Blend* of the images.

# Decide if the image below correctly follows the prompt:

## a photo of a suitcase left of a banana

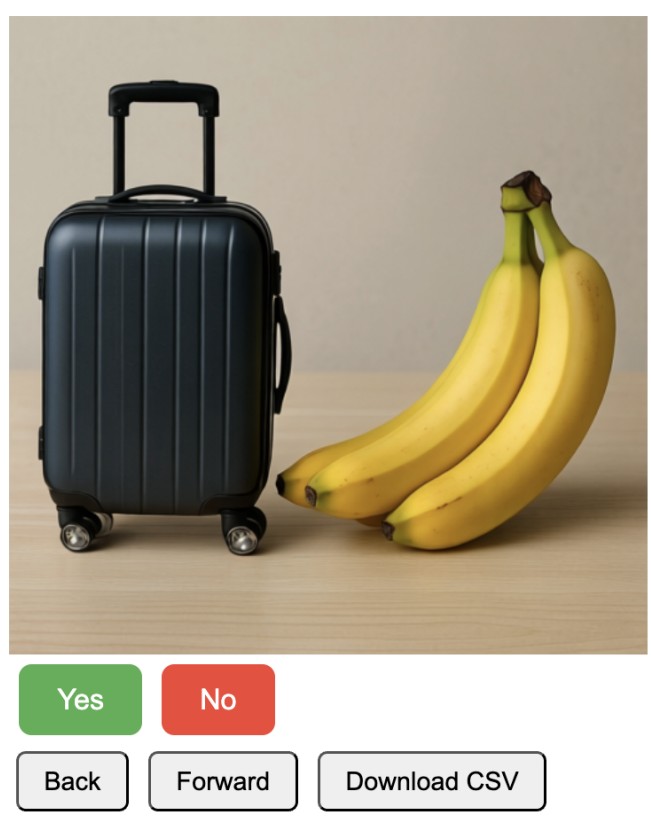

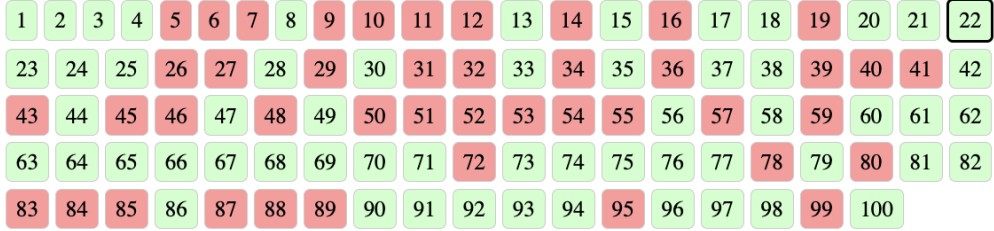

**Jump to Question:**

**Remaining Unanswered Questions:**

Figure 15: *PosEval* user study.

# Does the image look visually coherent (and not like many images put side by side)?

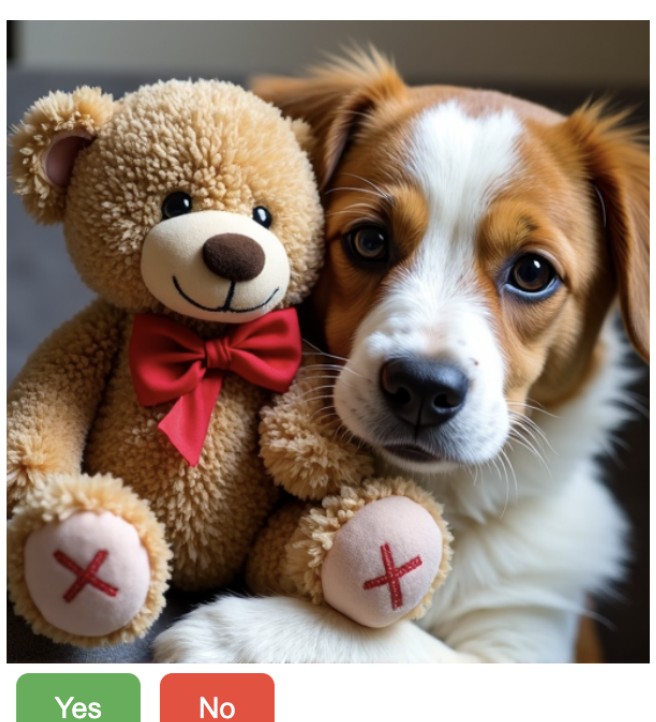

Yes    No

Back    Forward    Download CSV

**Jump to Question:**

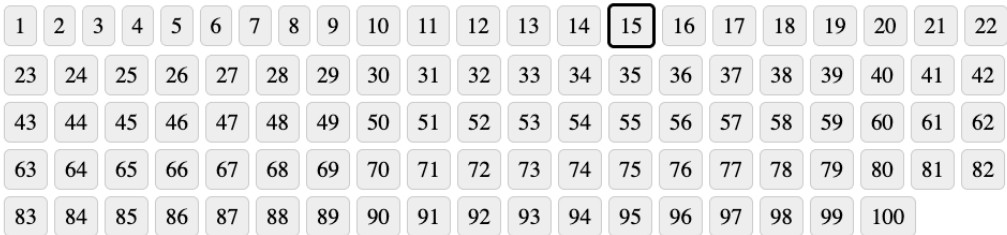

**Remaining Unanswered Questions:**

1, 2, 3, 4, 5, 6, 7, 8, 9, 10, 11, 12, 13, 14, 15, 16, 17, 18, 19, 20, 21, 22, 23, 24, 25, 26, 27, 28, 29, 30, 31, 32, 33, 34, 35, 36, 37, 38, 39, 40, 41, 42, 43, 44, 45, 46, 47, 48, 49, 50, 51, 52, 53, 54, 55, 56, 57, 58, 59, 60, 61, 62, 63, 64, 65, 66, 67, 68, 69, 70, 71, 72, 73, 74, 75, 76, 77, 78, 79, 80, 81, 82, 83, 84, 85, 86, 87, 88, 89, 90, 91, 92, 93, 94, 95, 96, 97, 98, 99, 100

Figure 16: *Blend* user study.

# I OVERLAPPING REGIONS

In Figure 17, we illustrate how Cutout enables Stitch to handle overlapping bounding boxes, allowing it to generate objects *in front of* or *behind* one another. This also equipts Stitch with the flexibility to handle various MLLM-generated bounding boxes, including those that may overlap.

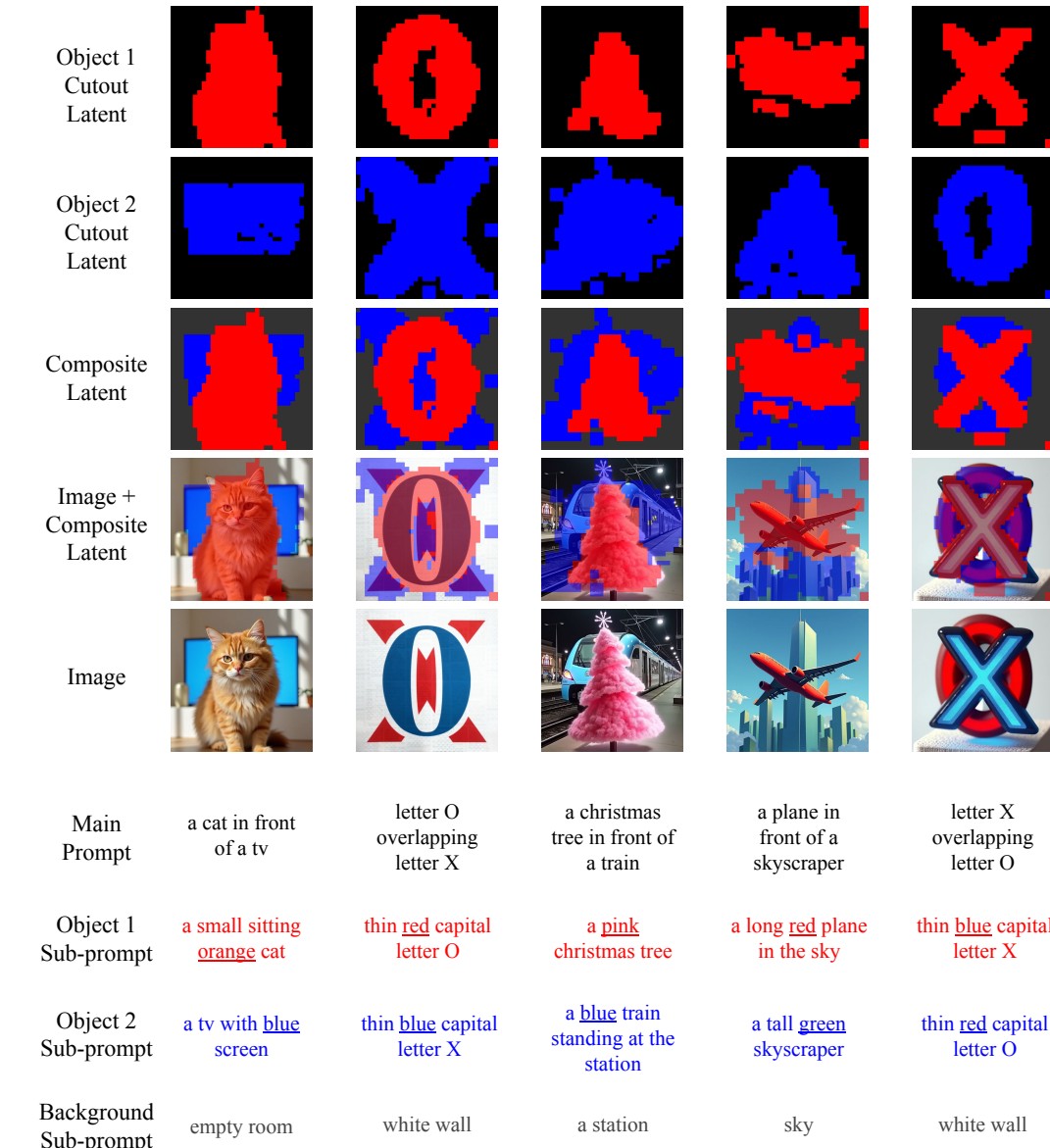

| | | | | | |
|---|---|---|---|---|---|
| **Main Prompt** | a cat in front of a tv | letter O overlapping letter X | a christmas tree in front of a train | a plane in front of a skyscraper | letter X overlapping letter O |
| **Object 1 Sub-prompt** | a small sitting orange cat | thin red capital letter O | a pink christmas tree | a long red plane in the sky | thin blue capital letter X |
| **Object 2 Sub-prompt** | a tv with blue screen | thin blue capital letter X | a blue train standing at the station | a tall green skyscraper | thin red capital letter O |
| **Background Sub-prompt** | empty room | white wall | a station | sky | white wall |

Figure 17: Stitch's use of Cutout enables it to handle overlapping bounding boxes. In an extreme setting, we task Stitch with generating pairs of objects that share identical bounding boxes spanning the entire image. By stacking the Cutout object latents in a controlled order (allowing partial occlusion), Stitch can render both objects while preserving the correct relational structure. Notably, the generated objects appear within regions specified by the cutout masks. Moreover, objects can retain underlined attributes from their sub-prompts even when those attributes are absent from the main prompt, providing further evidence that Stitch successfully merges overlapping sub-generations. This is not achievable with earlier methods operating strictly on disjoint rectangular bounding boxes.

## J  VALIDATION OF CUTOUT HEAD SELECTION

To validate the selected Cutout head's IoU, we generate a second set of 80 objects. To ensure broad and balanced coverage of real-world concepts, we define eight high-level categories and sample ten representative words from each, curated by ChatGPT (OpenAI, 2025) and non-overlapping with the GenEval objects. Prompts are made as, "A photo of a {obj}". Each prompt is repeated for 3 seeds.

Table 9: Validation of FLUX's best Cutout head performed on 80 prompts across 10 categories. Results are consistent across categories and closely match those observed during head selection.

|  | IoU | IoT |
|---|---|---|
| Animals | 0.72 | 0.91 |
| Clothes | 0.71 | 0.87 |
| Electronics | 0.58 | 0.90 |
| Food | 0.65 | 0.93 |
| Household | 0.66 | 0.90 |
| Nature | 0.59 | 0.89 |
| Plants | 0.56 | 0.89 |
| Transportation | 0.55 | 0.89 |
| All | $0.63_{\pm 0.06}$ | $0.90_{\pm 0.02}$ |

We evaluate our selected FLUX head (Block 14, Head 20) using its best-performing value of $\eta = 0.75$ on the new prompts. The results, shown in Table 9, yield 0.63 IoU and 0.90 IoT, closely matching the metrics observed during head selection (0.62 IoU and 0.92 IoT). The standard deviation across categories is also small, with the IoT being particularly consistent at only $\pm 0.02$.

The words chosen are:

*Animals:* hamster, monkey, turtle, lion, tiger, kangaroo, panda, gorilla, hippopotamus, penguin
*Clothes:* shirt, jacket, coat, dress, skirt, jeans, sock, shoe, hat, scarf
*Food:* strawberry, grapes, potato, tomato, loaf of bread, slice of cheese, milk, egg, yogurt, pasta
*Household Objects:* pillow, blanket, mug, coffee maker, trash, toy, candle, basket, kettle, plate
*Nature:* rock, leaf, branch, shell, feather, pinecone, volcano, mountain, acorn, iceberg
*Plants:* sunflower, cactus, pine, bamboo, oak tree, wheat, rose, daffodil, tomato plant, tulip
*Technology:* tablet, smartwatch, computer mouse, earbuds, camera, printer, camcorder, scanner, gaming console, speaker
*Transportation:* van, tram, helicopter, sailboat, cruise ship, tractor, scooter, jet, hot air balloon, roller blades

## K IMPACT OF CUTOUT HEAD SELECTION ON STITCH

To better understand how the chosen attention head for Cutout influences Stitch, we examine three questions: (1) what the distribution of IoU scores looks like across attention heads, (2) how well our proposed head-scoring method correlates with Stitch performance, and (3) how sensitive Stitch is to the specific Cutout head used. Our results show that although many heads are unsuitable for Cutout, several can achieve similarly high performance, and that our proposed SAM IoU-based method can effectively select these heads. While selecting the Cutout head with our IoU method only achieves most of the optimal performance, we also propose a validation-based method that can further boost performance.

**Distribution of IoU scores:** We analyze the distribution of IoU values across all FLUX heads, shown in the histogram in Figure 18. Most heads achieve similarly low IoU, with only a small number standing out as above average. This is intuitive: only heads that meaningfully attend to the foreground object are expected to yield higher IoU, and relatively few heads in the model appear to play this role.

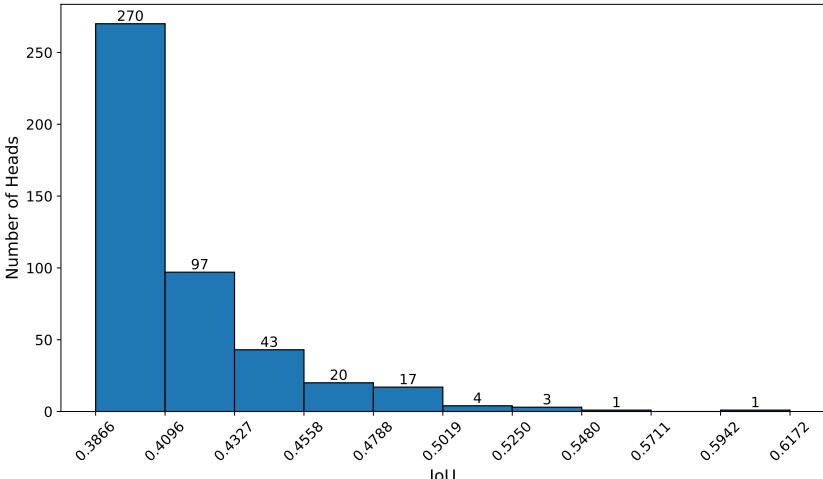

Figure 18: Histogram of the IoU of all FLUX heads when used for Cutout. Multiple attention heads are suitable for Cutout, but most are not.

**Correlation with Stitch performance:** Next, we verify that the IoU between attention maps and SAM masks is meaningfully correlated with Stitch's performance, specifically with *Blend*. We evaluate Stitch + FLUX on PosEval using Cutout derived from several different attention heads. Testing all $19 \times 24$ attention heads in FLUX would be infeasible, so we select a representative subset. To do this, we first rank all attention heads by their highest IoU with SAM masks across $\eta \in 0.75, 0.80, 0.85, 0.90, 0.95, 0.97, 0.99$. From this ranking, we select the top five heads as alternatives to the best-performing head used in Stitch. For additional comparison, we also include the three heads with median performance and the three lowest-performing heads.

Table 11 shows that selecting heads based on their IoU with SAM masks is effective: only the top heads achieve both high accuracy and high *Blend*. Heads with low IoU do not reliably focus on foreground objects. As a result, they may either retain enough of the object to preserve accuracy while keeping too much background and hurting *Blend*, or they may remove too much of the object, reducing accuracy altogether (e.g., Block 13 Head 10).

**Sensitivity of Stitch to Cutout head:** Next, we examine how sensitive Stitch is to the specific Cutout head used. The results in the *Middle* and *Bottom Head* groups show that randomly choosing a head is ineffective, as most heads are not well suited for this task. However, within the *Top Heads* group, several options perform similarly. In particular, the three highest-ranked heads (Block 14 Head 20, Block 17 Head 23, and Block 13 Head 21) achieve comparable accuracy and *Blend* scores.

Table 10: Stitch + FLUX performance on PosEval for by Cutout head. Heads are ranked by their best IoU across a range of thresholds, and we evaluate the top five, middle three, and bottom three. Only the top heads achieve both high accuracy and high *Blend*, demonstrating that our selection strategy is closely aligned with Stitch's performance. Several top heads perform similarly, meaning Stitch's effectiveness does not hinge on a single head. Red highlights poor performance.

| Rank | Block | Head | IoU | 2 Obj | 3 Obj | 4 Obj | Neg | Rel | PAB | Blend |
|------|-------|------|-----|-------|-------|-------|-----|-----|-----|-------|
| *Without Cutout* | | | | | | | | | | |
| | | | | 0.81 | 0.58 | 0.52 | 0.90 | 0.63 | 0.51 | 0.68 |
| *Top Heads* | | | | | | | | | | |
| 1 | 14 | 20 | 0.62 | 0.70 | 0.44 | 0.38 | 0.83 | 0.48 | 0.44 | 0.95 |
| 2 | 17 | 23 | 0.56 | 0.69 | 0.47 | 0.41 | 0.84 | 0.51 | 0.47 | 0.95 |
| 3 | 13 | 21 | 0.54 | 0.72 | 0.51 | 0.48 | 0.86 | 0.57 | 0.51 | 0.98 |
| 4 | 14 | 15 | 0.54 | 0.74 | 0.52 | 0.40 | 0.86 | 0.53 | 0.49 | 0.88 |
| 5 | 10 | 1 | 0.53 | 0.79 | 0.51 | 0.50 | 0.86 | 0.57 | 0.47 | 0.90 |
| *Middle Heads* | | | | | | | | | | |
| 227 | 15 | 19 | 0.40 | 0.74 | 0.52 | 0.46 | 0.84 | 0.53 | 0.47 | 0.76 |
| 228 | 12 | 14 | 0.40 | 0.77 | 0.56 | 0.52 | 0.87 | 0.58 | 0.51 | 0.73 |
| 229 | 6 | 21 | 0.40 | 0.76 | 0.56 | 0.52 | 0.88 | 0.52 | 0.51 | 0.83 |
| *Bottom Heads* | | | | | | | | | | |
| 454 | 3 | 17 | 0.39 | 0.78 | 0.52 | 0.51 | 0.87 | 0.55 | 0.48 | 0.68 |
| 455 | 1 | 11 | 0.39 | 0.70 | 0.48 | 0.51 | 0.87 | 0.36 | 0.41 | 0.83 |
| 456 | 13 | 10 | 0.39 | 0.34 | 0.18 | 0.13 | 0.62 | 0.20 | 0.15 | 0.90 |

**Advanced head selection strategy**  Because both the SAM masks used for head selection and the generation process itself introduce noise, the optimal head may lie within the top $k$ candidates rather than being the single highest-ranked one. Indeed, we observe that the third-ranked head, Block 13 Head 21, performs slightly better than the top two. If marginally higher accuracy is desired, a small validation set of Stitch prompts can be used to choose the best head from among the top $k$. Nevertheless, we emphasize that selecting just the top head ($k = 1$) already yields strong performance. As Figure 18 shows that very few heads achieve high IoU, even a very small $k$ is sufficient.

## L    IMPACT OF VARYING $\eta$ ON MASK QUALITY

We investigate the impact of $\eta$ on Cutout mask quality, focusing on two questions: (1) how IoU and IoT between Cutout masks and SAM masks vary with $\eta$, and (2) how these changes affect Stitch performance. Our results show that a wide range of $\eta$ values can capture most of the improvements over the base model, including the values proposed in the main paper. Additionally, we outline a strategy for further boosting accuracy with more vigorous hyperparameter search.

**IoU dependance on $\eta$:**    Figure 19 shows IoU and IoT for varying $\eta$ values on FLUX's 10 highest IoU heads. Together, the plots show that top-performing heads (Block 14 Head 20 and Block 17 Head 23) consistently focus on foreground objects, maintaining highest IoU across all thresholds, while increasing $\eta$ boosts IoT for all heads. This suggests that once the optimal head is identified, we can raise $\eta$ to avoid missing object parts while still excluding much irrelevant background. Moreover, several other heads also score highly, indicating multiple heads perform well.

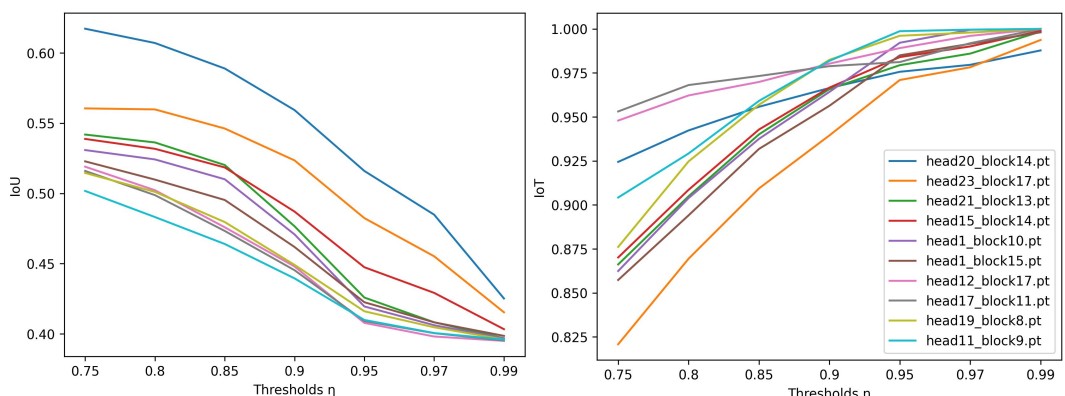

Figure 19: IoU and IoT between Cutout and SAM masks.

$\eta$ **impact on Stitch:**    Next, we examine how changing $\eta$ affects Stitch's performance. Using Cutout with the best FLUX head, Block 14 Head 20, we evaluate PosEval across different $\eta$ thresholds, as shown in Table 11. We observe that increasing $\eta$ improves performance on PosEval, whereas *Blend* performance declines.

Much of Stitch's gain over base models can be achieved with many $\eta$s. To maximize performance, it is advisable to first select the Cutout head and then increase $\eta$ as much as possible without causing a significant drop in *Blend* performance. However, the threshold search does not need to be very fine-grained; in this case, values between $0.90$ and $0.97$ yield reasonable results.

Table 11: Stitch + FLUX on PosEval using Cutout from Block 14 Head 20 across different $\eta$s. *Base* represents FLUX out-of-the-box.

| $\eta$ | 2 Obj | 3 Obj | 4 Obj | Neg | Rel | PAB | Blend |
|------|-------|-------|-------|------|------|------|-------|
| Base | 0.22 | 0.06 | 0.02 | 0.62 | 0.03 | 0.15 | 1.00 |
| 0.75 | 0.51 | 0.30 | 0.18 | 0.71 | 0.31 | 0.29 | 0.99 |
| 0.80 | 0.51 | 0.36 | 0.22 | 0.75 | 0.35 | 0.35 | 0.98 |
| 0.85 | 0.59 | 0.40 | 0.24 | 0.80 | 0.38 | 0.39 | 1.00 |
| 0.90 | 0.67 | 0.42 | 0.31 | 0.82 | 0.47 | 0.42 | 1.00 |
| 0.95 | 0.70 | 0.44 | 0.38 | 0.83 | 0.48 | 0.44 | 0.95 |
| 0.97 | 0.75 | 0.52 | 0.47 | 0.87 | 0.51 | 0.49 | 0.95 |
| 0.99 | 0.79 | 0.56 | 0.59 | 0.89 | 0.57 | 0.51 | 0.80 |

# M ALTERNATIVES TO HEAD SELECTION BASED ON SAM

To select a head for Cutout, we primarily recommend comparing the potential Cutout masks from each attention head with SAM maps, as described in Section 5.2. However, there are also several potential methods for selecting a head that do not require any external models.

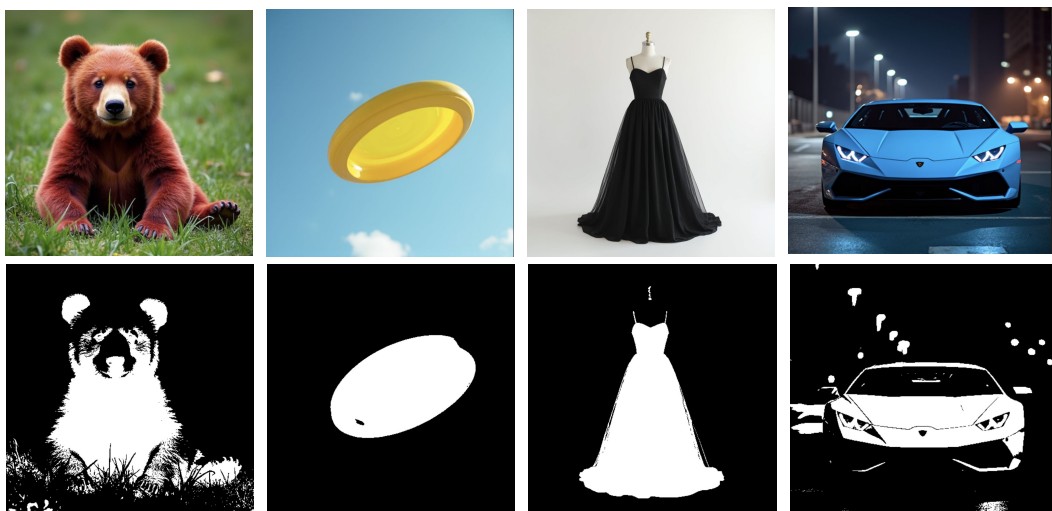

Figure 20: Masks generated by classifying each pixel based on whether its color is closer to that of the central pixel or to the average color of pixels along the side edges. With enough prompts featuring objects in diverse and clearly distinct colors, this method could serve as an alternative to SAM for finding ground truth to select the appropriate cutout head.

The first alternative is most similar to our original method, but the masks are generated without an external model. Images are first produced with distinct colors differentiating foreground and background objects, as illustrated in the top row of Figure 20. Masks are then created using a simple heuristic: each pixel is classified based on whether its color is closer to the central pixel (assumed to belong to the foreground) or to the average color of the side edge pixels. The bottom row of Figure 20 illustrates examples of these masks. Masks generated in this way replace the SAM masks described in Section 5.2, while the remainder of the selection process remains unchanged.

Our second head selection alternative takes a different approach. We begin with the assumption that a head focusing on foreground objects will primarily attend to the center of the image. To capture this, we model the expected average attention of a foreground-focused head as a 2D Gaussian centered in the latent space, with width $\sigma = \frac{L_s}{6}$, where $L_s = 32$ is the latent width. For each attention head, we compute its average attention map over the same 80 prompts used in our original selection strategy based on GenEval objects. We then compare these average attention maps with the described Gaussian using Jensen-Shannon (JS) divergence. Ranking heads by JS divergence reveals that those highly ranked by our original metric are also ranked highly by this method, suggesting that JS divergence could serve as a practical, model-free alternative to IoU. Figure 21 shows a histogram of the JS divergences of FLUX's attention heads, along with examples of the average attention maps for selected heads at different JS divergence values.

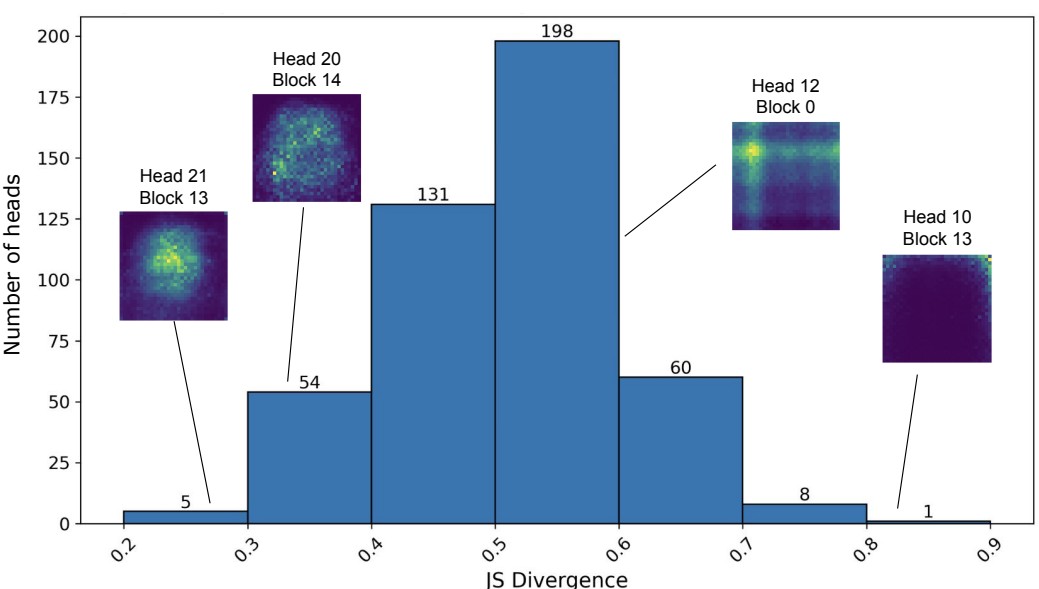

Figure 21: A histogram of Jensen–Shannon (JS) divergences between each head's average attention-map pattern and a centered 2D Gaussian with $\sigma \approx L_S/6$, where $L_s$ is the attention-map size. Heads with the lowest divergences (e.g. Head 21 in Block 13 and Head 20 in Block 14) closely match the expected Gaussian-like pattern and correspond to the top-performing heads based on IoU with SAM-derived ground truths. In contrast, heads with the highest divergences (e.g. Head 12 in Block 0 and Head 10 in Block 13) display off-center attention, indicating that they fail to concentrate on the foreground object.

## N    SEMANTIC LEAKAGE IN OTHER METHODS

Two contemporaneous works, RAG (Chen et al., 2025c) and Regional Prompting (Chen et al., 2024a), also propose test-time modifications to MM-DiT models to enhance positional understanding by generating and combining sub-prompts. As shown in Table 1, Stitch outperforms these methods on PosEval. To investigate why, we present several qualitative examples in Figure 23. We observe that both Regional Prompting and RAG are more prone to merging multiple objects (e.g., "a photo of an elephant below a horse") or omitting objects entirely (e.g., "a photo of a bus above a boat"), which may explain their lower performance.

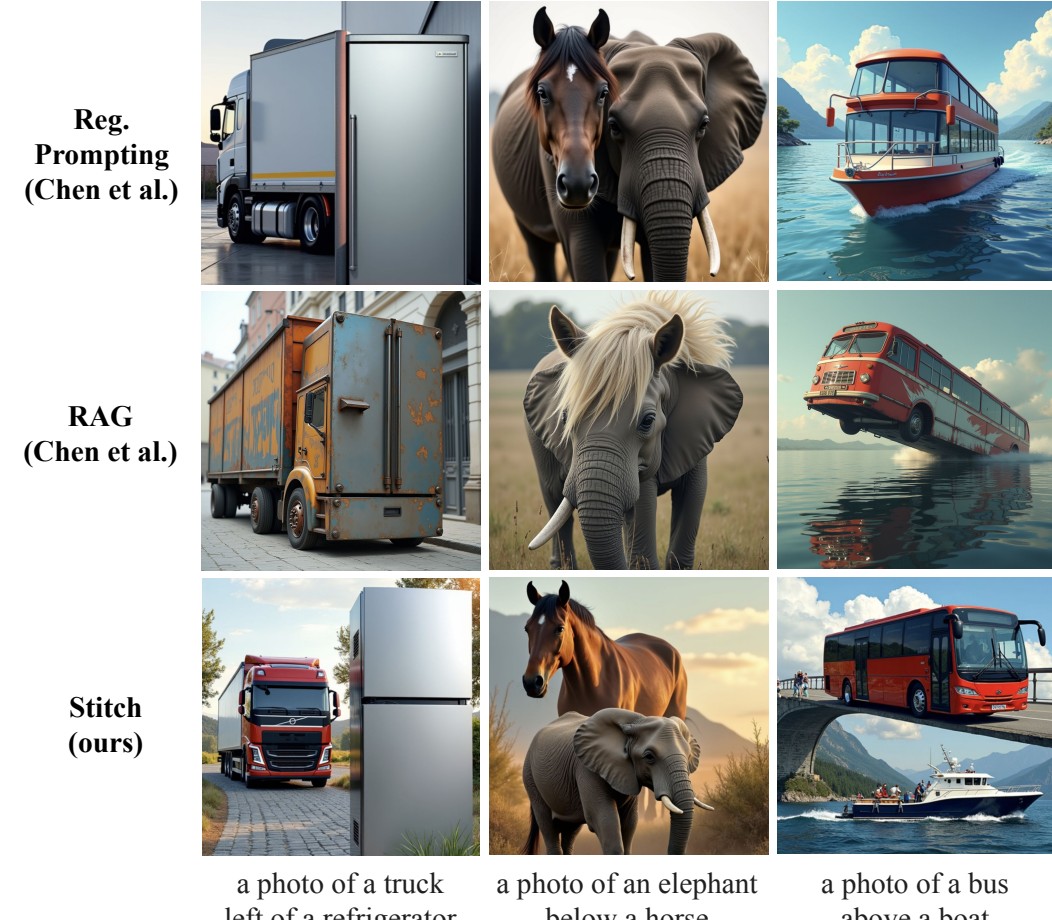

|  |  |  |
| --- | --- | --- |
| a photo of a truck left of a refrigerator | a photo of an elephant below a horse | a photo of a bus above a boat |

Figure 22: Reg. Prompting (Chen et al., 2024a) and RAG (Chen et al., 2025c) achieve lower scores than Stitch in Table 1, partly because their strategies for avoiding blending artifacts while constraining generation can lead to semantic leakage between regions. We illustrate such behavior across three different prompts, with all methods applied to FLUX.

## O   SENSITIVITY OF STITCH TO THE LLM-GENERATED BOUNDING BOXES

Because Stitch relies heavily on MLLM-generated bounding boxes, we evaluate its robustness to variations in these boxes. To do so, we conduct two experiments: perturbing the bounding boxes and shuffling them. We compare Stitch 's performance under these modifications to its performance with the original boxes, finding that Stitch is robust to variations in the bounding boxes, but strongly reliant on their accuracy. We use FLUX as the base model and evaluate on PosEval.

In the perturbation experiment, we examine how sensitive Stitch is to small, relation-preserving fluctuations in the bounding boxes. We perturb the boxes from the Original evaluation by randomly shifting their edges by up to $10\%$ of their initial width, ensuring that the underlying spatial relations remain correct. As shown in Table 12, performance under perturbation closely matches that of the original boxes, indicating that precise box placement is not required—small deviations do not meaningfully impact accuracy.

In the shuffling experiment, we assess how Stitch responds to incorrect bounding boxes. We keep the box locations fixed but randomly reassign the sub-prompts among them, producing many incorrect layouts. As shown in Table 12, this substantially degrades performance, indicating that Stitch relies strongly on the information encoded in the bounding boxes and is sensitive to incorrect assignments. Fortunately, LLMs are reliable at producing layout-aware sub-prompts that preserve these relations.

Table 12: Stitch + FLUX's PosEval performance under three conditions: the *Original* bounding boxes, *Perturbed* bounding boxes (always preserving correctness) and *Shuffled* sub-prompt/bounding box assignment (sometimes breaking correctness). Relative to the *Original* bounding boxes, Stitch maintains high accuracy with *Perturbed* bounding boxes, indicating robustness to small variations. In contrast, shuffling significantly impacts performance, showing that Stitch prioritizes the information encoded in the bounding boxes.

|           | 2 Obj | 3 Obj | 4 Obj | Neg  | Rel  | PAB  |
|-----------|-------|-------|-------|------|------|------|
| Original  | 0.70  | 0.44  | 0.38  | 0.83 | 0.48 | 0.44 |
| Perturbed | 0.71  | 0.42  | 0.40  | 0.82 | 0.45 | 0.44 |
| Shuffled  | 0.39  | 0.07  | 0.02  | 0.47 | 0.12 | 0.25 |

We also explore qualitatively how Stitch behaves when the bounding boxes are incorrect. We identify one failure case for "a photo of a dining table above a suitcase", shown in Figure 23: the MLLM mistakenly includes both objects in both sub-prompts instead of assigning one object to each. As a result, the table appears twice in the final image—once in the top region and once in the bottom.

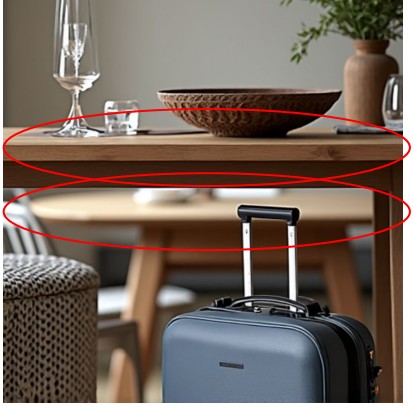

a photo of a dining table, wooden, tabletop and legs visible, viewed from the side, above a suitcase

a photo of a suitcase, wheeled, closed, handle visible, below a dining table

Figure 23: Erroneous MLLM-generated bounding boxes place both objects in each sub-prompt (right), resulting in the table being rendered twice (left).

