# OpenReview forum: "Stitch: Training-Free Position Control in Multimodal Diffusion Transformers"
_ICLR.cc/2026/Conference — Submitted to ICLR 2026_

### Official Review · Reviewer_RK61 · 2025-10-16

**Soundness:** 3
**Presentation:** 2
**Contribution:** 2
**Rating:** 4
**Confidence:** 5

**Summary:**

The paper proposes STITCH, a training-free positional control method for MMDiT T2I models. An external MLLM decomposes the prompt and predicts bounding boxes. The model first constrains attention inside each box (“Region Binding”), then uses one attention head to extract object latents (“Cutout”), stitches them with the background, and finishes generation.

**Strengths:**

- Works with off-the-shelf MMDiT/flow models and needs no finetuning.
- Clear mechanism (box-guided masking + attention-based Cutout).
- PosEval adds harder position tasks.

**Weaknesses:**

- Not end-to-end. It depends on an external MLLM for layout planning; the core model does not learn position.
- Novelty is modest. Bounding-box guidance, stitching, and attention-head probing are well-trodden ideas; the paper mainly adapts them to MMDiT.
- Sensitivity and brittleness are underexplored. Results may hinge on box quality, the chosen attention head, S steps, and η thresholds. Automatic head selection at test time is unclear.
- The approach adds test-time cost (generate per object for S steps), and the blend/accuracy trade-off suggests practical tuning overhead.
- Competent engineering with a useful benchmark, but the method is not end-to-end and the main ideas are incremental. PosEval has value, yet it does not offset the limited novelty of the core approach.

**Questions:**

What are the time/memory costs per image vs. base models and other training-free layout methods

---

> ### Author Response · Authors · 2025-11-19
>
> **(Q1) Memory and time cost**
>
> *a) Efficient implementation of Stitch*
>
> Because the objects are meant to occupy distinct parts of the image, the per-object predictions in the first $S$ steps can be obtained from a single network inference on a shared latent, using the union of all Region Binding constraints. Only the background needs its own separate inference, as the decision about which parts of the latent correspond to background is deferred until step $S$. After that, for the remaining $T − S$ steps, the model processes only a single composite latent with default predictions. Thus, the total number of required inferences becomes: $2S + (T − S) = T + S$.
>
> *b) Time comparison with base models and other training-free methods*
>
> The generation time of diffusion (flow-matching) models is heavily dominated by the time spent on noise (velocity) predictions made by neural networks. Typical base models like SD3.5 or FLUX which serve as our baseline make $T$ of such predictions, while Stitch and other methods may increase this number by introducing separate sub-predictions for different regions. In the table below, we summarize the total number of predictions of different methods.
>
> | Method              | Number of Predictions                    |
> |---------------------|-------------------------------------------|
> | Base models         | $T$                                       |
> | Stitch (ours)       | $(2 \cdot S_{\text{stitch}}) + (T - S_{\text{stitch}})$ |
> | RAG [1]             | $(K + 1) \cdot T$                         |
> | Reg. Prompting [2]  | $(2 \cdot S_{\text{reg}}) + (T - S_{\text{reg}})$ |
> | RPG [3]             | $(K + 1) \cdot T$                         |
> | LMD [4]             | $(K + 1) \cdot T$                         |
>
> \* $S_{\text{stitch}} \approx N/5$, $S_{\text{reg}} \approx N/3$
>
> Assuming we are interested in $K=4$ objects and use standard $T=50$, it translates to the number of steps shown in the next table.
>
> | Method              | Number of Predictions |
> |---------------------|------------------------|
> | Base models         | 50                     |
> | Stitch (ours)       | 60                     |
> | RAG [1]             | 250                    |
> | Reg. Prompting [2]  | 66                     |
> | RPG [3]             | 250                    |
> | LMD [4]             | 250                    |
>
> Stitch is highly efficient, requiring less computation than other training-free layout methods and only a few more predictions than the base models.
>
> *c) Memory comparison with base models and other training-free methods*
>
>
> The primary source of increased memory usage in training-free layout methods, relative to the base models, is the need to store sub-predictions for multiple regions concurrently. While methods like Stitch and [2] combine two predictions, methods [1,3,4] may require combining up to $K+1$ predictions, where $K$ is the number of objects. This requires storing between $1$ and $K$ latents for later mixing.
>
> In practice, this cost is negligible, as storing model weights requires far more resources. Compared to the $1.2 \times 10^{10}$ parameters in FLUX, a single latent tensor contains $32 \times 32 \times 64 \approx 6.5 \times 10^{4}$ values, six orders of magnitude fewer.
>
> **(W1, W5) Not end-to-end. It depends on an external MLLM for layout planning; the core model does not learn position.**
>
> Indeed, our method does not involve learning, offering a low-cost way to enhance SOTA T2I models by integrating them with an MLLM in a fully automated pipeline. It introduces positional conditioning in a non-trivial way to a model never explicitly trained for it. Furthermore, Stitch works with not only MLLM-generated but also human-provided layouts, which is useful when precise placement is necessary. In this case, the pipeline simplifies to a single model.

---

> ### Author Response · Authors · 2025-11-19
>
> **(W2, W5) Novelty of contributions.**
>
> *a) Unseen Performance*
>
> We are the first to propose a method that outperforms previous models, both base and bounding box enhanced, by up to **37 percentage points** on the PosEval benchmark (see **Table 1**, Stitch + FLUX Avg.).  Our approach delivers significant improvements across **three** different base models, including **doubling** the performance of the base FLUX model. Remarkably, we achieve all of this without any additional training, using novel procedures that overcome challenges in bounding box-guided generation in MMDiT.
>
> *b) Non-trivial adaptation of old bounding boxes and stitching methods to MMDiT*
>
> While prior methods explored guiding diffusion models through attention modulation to condition on bounding boxes, they were designed for older U-Net based architectures with a clear cross-attention bottleneck [3,5,6]. MM-DiT lacks this bottleneck and naive attention constraints, applied uniformly across all layers and timesteps, are ineffective. They are either too weak to enforce the layout or too strong, preventing smooth blending. In contrast, we find that applying strong constraints only during the first few timesteps is both sufficient and effective.
>
> For previous works to achieve spatial control by stitching sub-generations, they needed to either combine the latents from (a) full bounding box regions which may contain conflicting background information and therefore create non-coherent images [1,2] or (b) tight object foregrounds acquired by fully generating each object separately and creating cutouts with SAM from the final images [4]. Stitch obtains tight foregrounds without these inefficiencies: it only requires sub-generations during a few initial steps, after which the trajectories are merged into a single generation. With efficient implementation, these sub-generations can even be processed in parallel. Moreover, our novel use of attention heads removes the need for SAM-based cutouts at test-time altogether.
>
> As Reviewer 8ZJA noted, concurrent work published at ICCV '25 [1] also tackles this challenge, underscoring the non-triviality of the problem. However, our extended experiments in **Table 1** demonstrate that their approach falls short: their control–blend tradeoff, achieved by mixing attention-constrained and unconstrained predictions, suffers from object leakage between bounding boxes (see **Appendix N: Semantic Leakage in Other Methods**), resulting in much lower PosEval performance (see **Table 1**, 27% vs ours 55%).
>
> We have also added a new paragraph to the **Introduction** to better clarify the novelty and motivation of our work.
>
> *c) New insight into attention mechanisms*
>
> While attention patterns have been studied, we believe our work reveals new findings. We show that a single attention head, as early as at timestep 10 out of 50, can reliably predict the position of the foreground object (see **Section 5.2: Assessing Foreground Segmentation Accuracy of Cutout**). We verify that this behavior is consistent across multiple models (FLUX, Qwen-Image, SD3.5). We also demonstrate an unconventional application of this insight: using the early attention signal in Stitch to efficiently perform cutouts before the objects are fully generated.

---

> ### Author Response · Authors · 2025-11-19
>
> **(W3) Sensitivity underexplored on box quality, the chosen attention head, S steps, and η thresholds. Test-time selection not clear**
>
> Thank you for your suggestions regarding where we could provide further insights into our method’s behavior. We have updated the manuscript to include additional ablations and analyses addressing the areas you highlighted.
>
> *a) Choosing attention head*
>
> We further validate the selected Cutout attention head by examining how well it generalizes to new prompts and across object categories (see **Figure 5** in **Section 5.2: Assessing Foreground Segmentation Accuracy of Cutout**, **Table 9** in **Appendix J: Validation of Cutout Head Selection**). To this end, we use 80 additional prompts, evenly divided among 8 high-level categories. We then evaluate the IoU and IoT of the Cutout masks from our chosen FLUX head against SAM masks, finding that the results closely match those from the original head-selection process and that performance is consistent across categories.
>
> We also provide additional analysis of the impact of the selected Cutout head (see **Appendix K: Impact of Cutout Head Selection on Stitch**). Our goal is to understand (1) the distribution of IoU scores across attention heads, (2) how well our head-scoring method correlates with Stitch performance, and (3) Stitch’s sensitivity to the specific Cutout head used. We begin by presenting a histogram of IoU values with SAM masks for all FLUX heads, showing that only a small subset of heads is potentially suitable for Cutout (see **Figure 18**). We then evaluate PosEval performance using 11 different attention heads: the five highest-IoU heads, three median heads, and the three lowest heads (see **Table 10**). These results indicate that only the top heads achieve both high accuracy and strong blend, demonstrating that our SAM IoU–based selection strategy correlates well with Stitch performance. Finally, among the top five heads, we find that the top three perform similarly, suggesting that Stitch’s performance does not hinge on a single specific head. We additionally refer to these results in **Section 5.2: Assessing Foreground Segmentation Accuracy of Cutout**.
>
> *b) Choosing threshold η*
>
> We include additional studies on the impact of η on Cutout mask quality in **Appendix L: Impact of η on Mask Quality**. Specifically, we examine (1) how IoU and IoT between Cutout masks and SAM masks vary with η, and (2) how these variations influence Stitch’s overall performance. Our results indicate that a wide range of η values can capture most of the improvements over the base model, including the values proposed in the main paper. First, we present plots of IoU and IoT for the top-10 FLUX attention heads (see **Figure 19**), showing that the top heads consistently focus on foreground objects across different η values; notably, IoT always increases. This behavior allows us to select the optimal Cutout head based on IoU while tuning η to balance accuracy and blend via IoT. We further demonstrate how varying η affects performance on PosEval for our chosen FLUX head (see **Table 11**), finding that multiple η values achieve similar performance. We additionally refer to these results in **Section 6: Additional Analysis and Ablations of Stitch**.
>
> *c) Choosing number of steps S*
>
> Regarding S, we include an ablation in **Figure 9** (see **Section 6: Additional Analysis and Ablations of Stitch**) that examines how modifying S affects accuracy and blend, and how this trade-off guides the selection of S for Stitch. From this analysis, we find that S = 10 is the largest value possible before Blend begins to drop significantly, making it a reasonable choice. If you find this experiment insufficient, we would be happy to consider an additional, more targeted evaluation of S.
>
> *d) Sensitivity to box quality*
>
> We investigate this in **Appendix O: Sensitivity of Stitch to the LLM-Generated Bounding Boxes**. Because Stitch relies heavily on the bounding boxes produced by the MLLM, we aim to understand both the aspects in which it is robust and the conditions under which it fails. To this end, we apply two modifications to the original boxes: (1) we perturb their positions by up to 10% to test Stitch’s tolerance to variation in box placement, and (2) we randomly permute the sub-prompts associated with the boxes—creating many incorrect layouts—to evaluate Stitch’s behavior when the layout itself is wrong.
>
> As shown in **Table 12**, Stitch’s PosEval results change very little under perturbations but degrade substantially under incorrect layouts. This indicates that Stitch is robust to fluctuations in bounding box location but prioritizes information in the spatial layout, meaning it must be correct for the method to succeed.
>
> We additionally refer to these results in **Section 6: Additional Analysis and Ablations of Stitch**.

---

> ### Author Response · Authors · 2025-11-19
>
> *e) Clarification on head selection*
>
> Head selection is performed once per model, so at test time we simply use the fixed, pre-identified head. As a result, for FLUX, Qwen-Image, and SD3.5, future work can directly use the heads we have identified without needing to repeat the selection procedure. We have added a clarifying sentence at the end of the first paragraph in **Section 5.2: Assessing Foreground Segmentation Accuracy of Cutout to make this more explicit**.
>
> **(W4) Test-time cost from S steps and tuning necessary from blend-accuracy trade-off**
>
> *a) Test-time cost of Stitch*
>
> We acknowledge the additional test-time steps introduced by Stitch compared to the base model. However, because many use cases depend on accurate prompt following, it is also important to consider effective efficiency: the expected number of steps required to obtain a correct image. As outlined above in Q1, Stitch can be implemented by duplicating generation once for $S$ steps, resulting in a total of $T + S$ steps, whereas the base model requires $T$ steps. Let $X_b$ denote the base model’s accuracy and $X_s$ Stitch’s accuracy. The expected number of generations needed to obtain a correct image is therefore $\frac{1}{X_b}$ for the base model and $\frac{1}{X_s}$ for Stitch, yielding expected compute costs of $\frac{T}{X_b}$ and $\frac{T + S}{X_s}$, respectively. Stitch becomes the more efficient option when
>
> $$X_s \geq X_b \cdot \frac{T + S}{T}.$$
>
> For FLUX and SD3.5, where $S = 10$ and $T = 50$, this condition becomes $X_s \geq 1.2 \cdot X_b$. For Qwen-Image, with $S = 6$ and $T = 50$, we require $X_s \geq 1.12 \cdot X_b$.
>
> In the main results (**Table 1**), we see that Stitch surpasses these thresholds across all datasets, indicating that—despite requiring more steps per generation—it produces the correct image in fewer expected steps on average.
>
> *b) Tuning overhead*
>
> It is true that Stitch has an accuracy-blend tradeoff. This is part of why we report results on three widely used, state-of-the-art models (FLUX, Qwen-Image, and SD3.5): future users can directly benefit from our optimization efforts. For those looking to adapt Stitch to other models, the process is fortunately straightforward, as S is typically small for these models, so locating a sufficient value generally requires only a modest search.

---

> ### Comment · Reviewer_RK61 · 2025-11-25
>
> I thank the authors for their detailed response and the considerable effort put into the additional experiments. I maintain my view that the core method, while effective, relies heavily on external MLLM guidance and represents a practical integration rather than a significant breakthrough. Acknowledging that the authors have partly resolved the specific weaknesses I initially highlighted, I have raised my rating to 6. Consequently, to reflect my mixed assessment regarding the strong empirical results versus the limited novelty, I have adjusted my confidence score to 2.

---

### Official Review · Reviewer_8ZJA · 2025-10-20

**Soundness:** 3
**Presentation:** 2
**Contribution:** 1
**Rating:** 2
**Confidence:** 4

**Summary:**

The authors address the problem of position control in Flow Matching Text-to-Image (T2I) models. They propose a new benchmark, POSEVAL, to evaluate incorrect object placement, extending current benchmarks that only assess a narrow aspect of the problem (e.g., only two items, no negations). The authors also propose a method to mitigate this issue. Their approach uses a Multimodal Large Language Model (MLLM) to decompose the full prompt into sub-prompts and corresponding bounding boxes. It then applies block attention within these regions to ensure correct item placement and refines the bounding boxes using specific attention heads found to be responsible for focusing on foreground objects to make a cuttoff and merge it into the final image. Finally, the authors provide quantitive evidence that their method is effective at improving positioning compared to vanilla T2I models anda few other baselines.

**Strengths:**

The authors address a genuine and important problem, providing both a new benchmark for evaluation and a new method to solve this problem.

**Weaknesses:**

The main weakness of the paper is that the authors overlook previous works that have proposed similar methods to solve this problem, such as [1, 2]. I would expect the authors to include these methods in their experiments and discuss how their proposed method differs from them.

[1] Region-aware text-to-image generation via hard binding and soft refinement (Chen et al.)

[2] Training-free regional prompting for diffusion transformers (Chen et al.)

**Questions:**

1. Why are you using an MLLM for the first step? Is there any use for the multimodality of the output?
2. Which MLLM (L) was used in your experiments?

Additional Suggestions:
1. You discussed how you selected the attention heads responsible for positioning foreground items by using 80 images with the same prompt template. I find this discovery interesting, but it is not well-evaluated. After choosing the attention heads, I would expect an evaluation on an additional, different set of prompts and seeds to ensure that these heads generalize in their role.

2. Your introduction cites only one work (Ghosh et al., 2023). Please make sure to discuss previous works in this section, explain how your method differs, clarify why previous methods could not be used for this problem, and add the appropriate citations.

---

> ### Author Response · Authors · 2025-11-19
>
> **(W1, AS2) Need for comparison to works [1,2]. Elaborate on previous works in the introduction.**
>
> Thank you for bringing works [1,2] to our attention. We compared our approach with theirs and discussed the differences both here and in the updated manuscript (see **Introduction, Related Works** and **Table 1** in **Main Experiments**).
>
> *a) Overview of Key Distinctions Relative to Methods [1] and [2]*
>
> First, we provide a quantitative comparison of Stitch with [1] and [2]. In the revised manuscript, we expand **Table 1** to include [1,2], using their publicly available code and recommended parameter settings. Both methods improve the baseline FLUX performance on PosEval (FLUX: 0.18, [1]: 0.27, [2]: 0.31), but still fall short of Stitch, which reaches 0.55 when applied to FLUX. We attribute this gap to the way these methods achieve blending, through soft mixing of latent predictions with and without locality constraints (see details in Sections b and c below). While effective for smooth blending, this approach risks positional leakage across regions, as illustrated in **Figure 22** (see **Appendix N: Semantic Leakage in Other Methods**).
>
> Compared to these contemporaneous methods, a key innovation in Stitch is its Cutout mechanism, which enables the extraction of foreground objects directly from the latents. In contrast, [1] and [2] are limited to merging larger regions that often contain conflicting background information from each sub-prompt. This capability is likely what allows Stitch to better suppress visible boundaries without introducing unwanted image changes that compromise prompt adherence in position-related prompts. These methods also differ in numerous implementation details, which are discussed in depth in Sections b and c below.
>
> We would also like to note that, in accordance with the ICLR submission guidelines, reference [1] (appearing in ICCV 2025) qualifies as contemporaneous work, while reference [2] is, to the best of our knowledge, available only as an arXiv preprint. While we appreciate the Reviewer allowing us to more clearly situate our contribution within the evolving literature, we kindly ask the Reviewer to consider these distinctions in the context of ICLR’s policy on contemporaneous and non-peer-reviewed work.
>
> *b) In-depth Theoretical Comparison with Method [1]*
>
> There are several similarities between [1] and Stitch, but key differences make them fundamentally distinct approaches. Both methods utilize latent predictions conditioned on separate sub-prompts within bounding boxes and combine them to form the final image. Both recognize that without careful combination, blending artifacts can emerge. However, the way each method addresses this challenge differs substantially.
>
> First, the methods diverge even in their preliminary stages, where sub-prompts are generated within their respective bounding boxes. Alongside these sub-prompt latents, each method also produces an additional latent. For Stitch, this extra latent comes from a background prompt and is generated independently during these steps. In contrast, [1] generates a latent using the full prompt. At each step, the latent from the full prompt is overwritten within the sub-prompt regions by the corresponding sub-prompt latents.
>
> After these preliminary steps, the methods diverge completely. In Stitch, we perform a single Cutout operation to extract and combine all latents, after which the remaining generation proceeds unconstrained. In contrast, [1] employs a procedure called regional soft replacement, which introduces a second set of highly descriptive, region-bound prompts—something Stitch does not need. At each step of this process, all predictions begin from the same latent. The primary latent is generated using the full prompt, while the sub-prompt latents are produced using their respective sub-prompts. Within each sub-prompt region, the primary latent is then mixed with the corresponding sub-prompt latent through weighted averaging, merging them back into a single latent before proceeding to the next step. Another key distinction is that Stitch can extract the foreground objects from the sub-prompt latents via Cutout, whereas [1] can merge content only across broad bounding box regions.

---

> ### Author Response · Authors · 2025-11-19
>
> *c) In-depth Theoretical Comparison with Method [2]*
>
> Similarly, [2] shares certain methodological elements with Stitch, but the two ultimately remain distinct. Both approaches decompose the full prompt into sub-prompts with associated bounding boxes, generate objects constrained to these boxes for a small number of steps S, and then complete the remaining generation without constraints. However, they already diverge within these constrained steps. In [2], each step begins from the same latent: the model produces the full-prompt and sub-prompt predictions separately and then re-mixes their latents via weighted averaging. Stitch, by contrast, generates the latents entirely independently for the S constrained steps and merges them only once, at step S.
>
> The combining mechanisms also differ. Whereas [2] performs weighted averaging between the sub-prompt and full-prompt latents within each token, Stitch selects between latents on a per-token basis. Moreover, Stitch can extract foreground objects using Cutout, while [2] is limited to merging across the full region.
>
> It is also important to note that [2] offers no quantitative assessment of the method’s accuracy, depending instead entirely on qualitative examples.
>
> **(AS1) Evaluation of Cutout head on an additional, different set of prompts and seeds.**
>
> We have included a more rigorous validation of the Cutout head in **Appendix J: Validation of Cutout Head Selection**. In this evaluation, we use 80 new prompts, evenly divided across eight high-level categories, and evaluate each on three seeds. In **Table 9**, we observe that IoU and IoT on this validation set (0.63 and 0.90, respectively) are very similar to the values obtained during Cutout head selection (0.62 and 0.92, reported in **Table 6**, **Appendix F: Top Cutout Segmentation Heads for Each Model**), indicating that the chosen head generalizes well to new prompts and seeds. We additionally refer to these results in **Section 5.2: Assessing Foreground Segmentation Accuracy of Cutout**.
>
> **(Q1) Why MLLM? Is there any use for multimodality?**
>
> As shown in **Table 1**, base T2I models exhibit difficulty generating accurate layouts. We address this by providing accurate bounding boxes, obtained either from users or automatically via a multimodal LLM (MLLM). We opt for a multimodal LLM rather than a unimodal LLM because the most advanced current LLMs are multimodal and trained on tasks such as visual question answering, which we expect to improve their scene understanding. Importantly, Stitch is compatible with any model capable of producing accurate text-to-layout outputs.
>
> **(Q2) Which MLLM (L) was used in your experiments?**
>
> We use GPT-5 to generate the object subprompts and bounding boxes, and GPT-4 to generate the background subprompt.

---

> > ### Comment · Reviewer_8ZJA · 2025-11-26
> > **Response to Authors**
> >
> > I thank the authors for their detailed response and the significant updates to the manuscript.
> >
> > The authors have addressed my primary concern regarding the comparison with previous works [1] and [2]. The inclusion of these methods in the main experiments (Table 1), along with the discussion on the theoretical distinctions, places the paper's contribution in the proper context. The quantitative results showing that Stitch significantly outperforms these baselines are convincing.
> >
> > Furthermore, I appreciate the additional experiments regarding the validation of the Cutout head selection.
> >
> > Finally, the clarifications regarding the choice of MLLM and the specific models used have answered my remaining questions.
> >
> > In light of these revisions and the thorough rebuttal, I have updated my score.

---

### Official Review · Reviewer_RAjF · 2025-10-30

**Soundness:** 3
**Presentation:** 3
**Contribution:** 3
**Rating:** 6
**Confidence:** 5

**Summary:**

This paper proposes STITCH, a training-free, test-time method to improve spatial relations in modern MMDiT/flow T2I models, and introduces PosEval, a harder position-focused benchmark extending GenEval  (3–4 objects, negative/relative relations, positional attribute binding).

**Strengths:**

1. Clear, practical contribution: a training-free procedure that plugs into state-of-the-art MMDiT/flow models and measurably improves spatial adherence.
2. Well-motivated attention masking (Region Binding): three simple constraints that isolate intra-box generation and reduce positional leakage across regions.
3. New benchmark adds breadth and difficulty: PosEval targets real failure modes (negatives, relative relations, scaling to 3/4 objects, attribute-position binding) and demonstrates that “Position is solved” is premature.

**Weaknesses:**

1. Occlusion/overlap and complex layouts: Bounding boxes are non-overlapping on a coarse grid. Many real prompts imply occlusions or partial overlaps; it’s unclear how well Cutout and Region Binding handle tight interactions or interpenetrating objects.
2. Head selection practicality: Selecting a single “good” attention head per base model uses a small calibration process (80 prompts with SAM-based IoU/IoT). While not training, it’s extra pre-deployment effort.

**Questions:**

1. How sensitive is Stitch to the quality of the LLM-generated bounding boxes and sub-prompts? Do you have failure analyses where LLMs produce impractical layouts or inconsistent object parsing?

2. Is there an automated, model-agnostic way to choose the Cutout head at test time without using SAM-based ground-truth masks? Could you use self-consistency or internal signals to pick heads dynamically?

3. How stable is the chosen head across prompts, resolutions, and seeds? Does a single head suffice for all object categories, or do some categories benefit from different heads?

4. Can Stitch handle overlapping or occluded objects (e.g., partial overlaps of boxes)? If not, what would be required to support this (e.g., layered depth ordering, soft box masks, or iterative compositing)?

---

> ### Author Response · Authors · 2025-11-19
>
> **(W1, Q4) Can Stitch handle overlapping or occluded objects?**
>
> Yes, Stitch has the ability to handle overlapping or occluded objects. Many existing bounding-box methods are only suitable for non-overlapping regions (a setting that is already challenging and relevant to many practical applications). However, an additional advantage of our Cutout mechanism beyond what was discussed in the main paper is that it enables the possibility to handle occlusions.
>
> As shown in **Figure 17** (see **Appendix I: Overlapping Regions**), we test an extreme setting in which Stitch generates two objects occupying identical bounding boxes that cover the entire image. By stacking the Cutout object latents in a controlled sequence, Stitch can render both objects with partial occlusion while preserving their relational structure. Objects appear within the regions specified by their Cutout masks and retain attributes from their sub-prompts that are absent in the main prompt. This provides evidence that objects emerge not just from corrections in later steps, but from the accurate merging of early latents.
>
> **(W2) Extra pre-deployment effort in choosing the Cutout head**
>
> We acknowledge the inconvenience of the required calibration process when selecting the Cutout head for a given model. Prior approaches were limited to combining rectangular bounding boxes [1, 2, 3], making them prone to blending artifacts from mismatched backgrounds, or required external segmentation models for each generation [4]. Our inexpensive, one-time-per-model calibration improves blending (see **Table 2**) with no need for external models at test-time, representing a step forward towards more practically useful methods.
>
> **(Q1) Sensitivity of Stitch to the quality of the LLM-generated bounding boxes. Failure analysis.**
>
> Stitch depends on the accuracy of the bounding boxes, but is otherwise robust to positional fluctuations. This indicates that, for Stitch to function correctly, bounding boxes do not need to be perfectly placed but only broadly accurate. We demonstrate this in **Table 12** (see **Appendix O: Sensitivity of Stitch to the LLM-generated Bounding Boxes**). We conduct two experiments: one in which we perturb the bounding boxes and another in which we shuffle object assignments between boxes. We additionally refer to these results in **Section 6: Additional Analysis and Ablations of Stitch**.
>
> In the perturbation experiment, we shift each bounding box by up to 10% to test whether Stitch tolerates variation in where the MLLM places the box. Performance remains nearly unchanged relative to the original boxes, indicating strong robustness to such fluctuations.
>
> In contrast, shuffling object assignments produces many incorrect layouts, and Stitch’s performance degrades substantially. This shows that Stitch largely prioritizes the positional information in the LLM-generated layout.
>
> Still, we have observed occasional LLM output failures. For example (see **Figure 23**, **Appendix O: Sensitivity of Stitch to the LLM-generated Bounding Boxes**), given the prompt *“a photo of a dining table above a suitcase,”* the LLM produced sub-prompts such as *“a photo of a dining table, wooden, tabletop and legs visible, viewed from the side, above a suitcase”* and *“a photo of a suitcase, wheeled, closed, handle visible, below a dining table.”* Instead of yielding two clean, independent descriptions—*“a photo of a dining table”* and *“a photo of a suitcase”*—the model embedded the spatial relations into each sub-prompt. This led Stitch to misinterpret the layout and generate two tables. Such errors could likely be eliminated with improved prompt design or through in-context learning.

---

> ### Author Response · Authors · 2025-11-19
>
> **(Q2) Is there a way to choose the Cutout head without using SAM?**
>
> Yes, there are several potential ways to select the Cutout head without relying on an external model. A detailed discussion of these methods is provided in **Appendix M: Alternatives to Head Selection Based on SAM**. The first approach is similar in spirit, but instead of using SAM to extract masks, we deliberately generate images that can be separated using simple foreground–background cues, specifically prompts specifying a foreground color distinct from the background (e.g., “a red bear in green grass”). Masks are then obtained by classifying each pixel as more similar either to the center pixel (representing the foreground object) or to the side edge (representing the background). This provides a model-free alternative to SAM masks. We include examples of masks generated using this method in **Figure 20**.
>
> The second alternative takes a completely different approach. Intuitively, attention heads that focus on foreground objects are likely to concentrate near the center of the image. We approximate this expected focus with a 2D Gaussian. For each head, we compute the average attention across 80 prompts and measure the similarity to the Gaussian using Jensen–Shannon (JS) divergence. We find that the top heads identified by this method largely overlap with those selected using our original approach. **Figure 21** shows a histogram of the JS divergences for the FLUX attention heads, along with examples of average attention maps for selected heads at different points in this distribution.
>
> **(Q3) Stability of the chosen head. Accuracy across many object categories.**
>
> We find that our selected head is relatively stable across different settings, as shown in **Table 9** (see **Appendix J: Validation of Cutout Head Selection**). In this experiment, we evaluate 80 new prompts, evenly divided into eight object categories, and measure the IoU and IoT of Stitch’s Cutout head with respect to SAM masks across three seeds per prompt. The results show that IoU and IoT on these new prompts closely match those observed on the original prompts (see **Table 6**, **Appendix F: Top Cutout Segmentation Heads for each Model**), indicating consistency across both prompts and seeds. Furthermore, the standard deviation across categories is low (±0.06 for IoU and ±0.02 for IoT), further demonstrating stability across object classes. We additionally refer to these results in **Section 5.2: Assessing Foreground Segmentation Accuracy of Cutout**.

---

> ### Author Response · Authors · 2025-11-27
>
> Dear Reviewer RAjF,
>
> As the discussion period deadline approaches, we would like to kindly ask whether you have any thoughts on the updated version of our manuscript and whether it has influenced your assessment. We would be happy to address any remaining questions or concerns.
>
> Best regards,
> The Authors

---

### Author Response · Authors · 2025-11-19

We sincerely thank the Reviewers for their thoughtful and encouraging feedback. They highlighted that our work is a *new method* (8ZJA) that makes a *clear, practical contribution: a training-free procedure that plugs into state-of-the-art MMDiT/flow models and measurably improves spatial adherence* (RAjF), and *needs no finetuning* (RK61). Reviewers also emphasized that our method Stitch uses a *clear mechanism (box-guided masking + attention-based Cutout)* (RK61), grounded in *well-motivated attention masking (Region Binding): three simple constraints that isolate intra-box generation and reduce positional leakage across regions* (RAjF). In addition, they recognized that we *address a genuine and important problem* (8ZJA).

Our PosEval benchmark was particularly valued: Reviewers noted that it is *new* (RAjF, 8ZJA), *useful* (RK61), *adds harder position tasks* (RK61), *breadth and difficulty, targets real failure modes (negatives, relative relations, scaling to 3/4 objects, attribute-position binding) and demonstrates that “Position is solved” is premature* (RAjF).

We have taken all reviewer comments into careful consideration and strengthened the manuscript based on their suggestions. We kindly ask the Reviewers to consider increasing their scores if they find our responses satisfactory. We have attached a revised version of the work with all changes highlighted in blue.


**References**

[1] Chen et al. Region-aware text-to-image generation via hard binding and soft refinement. ICCV, 2025.

[2] Chen et al. Training-free regional prompting for diffusion transformers. arXiv, 2024

[3] Yang et al. Mastering text-to-image diffusion: Recaptioning, planning, and generating with multimodal llms. ICML 2024.

[4] Lian et al. Llm-grounded diffusion: Enhancing prompt understanding of text-to-image diffusion models with large language models. TMLR 2024.

[5] Fang et al. Got: Unleashing reasoning capability of multimodal large language model for visual generation and editing. NeurIPS 2025.

[6] Feng et al. Layoutgpt: Compositional visual planning and generation with large language models. NeurIPS 2023.

---

### Author Response · Authors · 2025-12-02
**Summary of Discussion Period**

TL;DR: Days before the leak, scores raised to 6/6/6 following extensive additional experiments and clarifications (see blue in intro, experiments, and Appendix p.25-33).

We provide a summary of the discussion period, including that **two reviewers raised their scores to 6 (resulting in 6/6/6) several days before the leaked information became public**. We are grateful for the AC’s additional time and hope it is clear that all discussions and updates from both reviewers and authors were made in good faith.

Reviewer 8ZJA’s sole concern was the absence of comparisons to two specific works. We addressed this by adding the comparisons and demonstrating that our method outperforms them significantly (see Table below). This update led the reviewer to raise the score to 6. As noted in our rebuttal, **these works are contemporaneous** (ICCV25) and non–peer reviewed (arXiv). According to the ICLR reviewer guide (https://iclr.cc/Conferences/2026/ReviewerGuide), “Reviewers can make authors aware of related contemporaneous work or arxiv papers, but the lack of such comparisons cannot be a basis for rejection.” This score change was key, reflecting the reviewer’s honest assessment and **ensuring the review adheres to ICLR policy.**

Reviewer RK61 likewise increased their score to 6 after we provided extensive additional ablations and further discussion highlighting our method’s **unprecedented accuracy** (see Table below).

Below we include a condensed version of our primary results (Table 1), highlighting the results most relevant to the reviewers’ score changes. The benchmark evaluates accuracy across six positional tasks, where higher values indicate better performance.

| Model             | 2 Objects | 3 Objects | 4 Objects | Negative Relations | Relative Relations | Positional Attribute Binding | Avg. ↑ |
|------------------------------|-----------|-----------|-----------|---------------------|---------------------|------------------------------|--------|
| RPG [3] (requested by R-8ZJA)   | 0.28   | 0.07   | 0.01   | 0.65        | 0.06        | 0.14             | 0.20  |
| RAG [1] (requested by R-8ZJA)   | 0.50   | 0.11   | 0.04   | 0.61        | 0.13        | 0.21             | 0.27  |
| Reg. Prompting [2] | 0.25 | 0.17 | 0.28 | 0.68 | 0.30 | 0.19 | 0.31 |
| LMD [4]   | 0.74   | 0.43   | 0.23   | 0.54        | 0.35        | 0.46             | 0.46  |
| **Stitch (ours) + SD3.5 Large** | **0.53** | **0.22** | **0.12** | **0.79** | **0.27** | **0.37** | **0.38** |
| *Gain over SD3.5 Large*    | *+0.19* | *+0.16* | *+0.10* | *+0.15* | *+0.21* | *+0.21* | *+0.17* |
| **Stitch (ours) + FLUX.1 [Dev]** | **0.70** | **0.44** | **0.38** | **0.83** | **0.48** | **0.44** | **0.55** |
| *Gain over FLUX.1 [Dev]*    | *+0.48* | *+0.38* | *+0.36* | *+0.21* | *+0.45* | *+0.29* | *+0.37* |
| **Stitch (ours) + Qwen-Image**  | **0.87** | **0.67** | **0.61** | **0.93** | **0.43** | **0.77** | **0.71** |
| *Gain over Qwen-Image*     | *+0.11* | *+0.27* | *+0.40* | *+0.44* | *+0.33* | *+0.16* | *+0.28* |

References:

[1] Chen et al. Region-aware text-to-image generation via hard binding and soft refinement. ICCV, 2025.

[2] Chen et al. Training-free regional prompting for diffusion transformers. arXiv, 2024.

[3] Yang et al. Mastering text-to-image diffusion: Recaptioning, planning, and generating with multimodal llms. ICML 2024.

[4] Lian et al. Llm-grounded diffusion: Enhancing prompt understanding of text-to-image diffusion models with large language models. TMLR 2024.

---

### Meta-Review · Area_Chair_DH51 · 2026-01-07

**Summary:**

The initial ratings are 2, 6, 4. This paper proposes a benchmark, POSEVAL, to evaluate incorrect object placement, extending current benchmarks that only assess a narrow aspect of the problem (e.g., only two items, no negations).  Their approach uses a MLLM to decompose the full prompt into sub-prompts and corresponding bounding boxes. It then applies block attention within these regions to ensure correct item placement and refines the bounding boxes using specific attention heads found to be responsible for focusing on foreground objects to make a cuttoff and merge it into the final image.

Strengths:
(1) This paper introduces a training-free procedure that plugs into MMDiT/flow models and measurably improves spatial adherence.
(2) New benchmark adds breadth and difficulty: PosEval targets real failure modes (negatives, relative relations, scaling to 3/4 objects, attribute-position binding) and demonstrates that “Position is solved” is premature.


Weaknesses:
(1)Sensitivity and brittleness are underexplored. Results may hinge on box quality, the chosen attention head, S steps, and η thresholds. Automatic head selection at test time is unclear.
(2) The authors overlook previous popular works that have proposed similar methods to solve this problem.

**Reviewer Concerns:**

Some concerns of Reviewer RAjF were addressed by the rebuttal, and Some main concerns of  Reviewer 8ZJA and RK61 are still outstanding.

**Reviewer Scores:**

Reviewer 8ZJA could raise the rating score.

---

### Decision · Program_Chairs · 2026-01-26

Reject